# OASIS: Optimal Analysis-Specific Importance Sampling for event generation

**Konstantin T. Matchev and Prasanth Shyamsundar**[⋆]

Institute for Fundamental Theory, Physics Department, University of Florida,
Gainesville, FL 32611, USA

⋆ prasanths@ufl.edu

## Abstract

We propose a technique called Optimal Analysis-Specific Importance Sampling (OASIS) to reduce the number of simulated events required for a high-energy experimental analysis to reach a target sensitivity. We provide recipes to obtain the optimal sampling distributions which preferentially focus the event generation on the regions of phase space with high utility to the experimental analyses. OASIS leads to a conservation of resources at all stages of the Monte Carlo pipeline, including full-detector simulation, and is complementary to approaches which seek to speed-up the simulation pipeline.

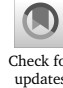

# 1 Introduction

## 1.1 Background

Numerical simulations play a key role in collider physics. Since typically there are no closed form expressions for the distribution of reconstructed collider events under various theory models, Monte Carlo (MC) methods [1] are needed to simulate the experimental outcomes predicted by competing theory models (or by different model parameters for the same theory model). Inferences about the underlying model and its parameters can then be made by comparing the observed data against the simulations. The sensitivity of such an inferential analysis depends both on the number of *real* events $N_r$ in the experimental data and on the number of *simulated* events $N_s$ available—the finiteness of the real and simulated samples introduces statistical uncertainties in the estimation of the true and theory-model distributions, respectively. In general, these uncertainties reduce with increasing volume of data; and while the available statistics for $N_r$ is determined by the integrated luminosity of the experiment, the amount of simulated data is in principle under our control. That is why it is desirable to have a sufficient volume of simulated data, i.e., $N_s \gg N_r$, so that the statistical uncertainty from its finiteness is not a dominant source of uncertainty in the analysis.

However, the pipeline for simulating reconstructed collider events is computationally expensive, and this poses an immense challenge to the high energy physics (HEP) community in terms of computational and storage resources. As a result, much too often experiments are forced to settle on an acceptable, but perhaps not ideal, value of $N_s$, which leads to additional limitations on the sensitivity [2]. This problem is expected to be exacerbated in the high-luminosity phase of the Large Hadron Collider (LHC) [3–7]. For this reason, many interesting ideas have been recently proposed to speed up individual stages of the simulation pipeline, as well as the entire pipeline [8–31]. The full detector simulation is the most resource-intensive aspect in this problem. In this paper we shall provide a parton-level optimization which, as we will discuss, would nevertheless result in resource conservation at all stages of the simulation pipeline. While motivated by the specific problem arising in HEP, we believe that our ideas are of more general mathematical interest and could potentially be usefully applied in other fields as well.

The first step of the simulation pipeline is the parton-level event generation [32, 33]. The distribution of parton-level events can be computed from the relevant matrix-elements and parton distribution functions (pdfs). However, it is not straightforward to sample events ac-

cording to a given multi-dimensional distribution, except in special cases (e.g., when the distribution is specifically engineered to be easy to sample from). This difficulty is typically handled as follows.

Let the multi-dimensional random variable $x$ represent a parton-level event and let $f(x)$ be the differential cross-section (unnormalized distribution) of $x$ for a given process under a given theory model. The integral of $f$ over the domain gives the cross-section of the process under consideration. With this backdrop, the goal of the MC simulation is twofold:

- *Event generation.* On the one hand, to obtain a sample of events distributed throughout the phase space as-per $f$:

$$\frac{dn}{dx} \propto f(x). \tag{1}$$

- *Cross-section estimation.* At the same time, to compute the integral of $f$

$$F \equiv \int dx \, f(x), \tag{2}$$

which can be used to properly normalize (1), e.g., to scale the simulations to match the available statistics in the real data.

In order to accomplish these goals, one makes use of an alternative normalized distribution $g(x)$, referred to as the sampling distribution, which is easy to sample from. Then, as an alternative to generating events as-per $f(x)$, we can generate events according to $g(x)$ and weight each sampled event $x$ by

$$w(x) \equiv \frac{f(x)}{g(x)}. \tag{3}$$

The mean weight of the sampled events can serve as an estimate of the integral $F$, since

$$F = \int dx \, f(x) \frac{g(x)}{g(x)} = \int dx \, g(x) \, w(x) \equiv E_g[w(x)], \tag{4}$$

where $E_g[\cdots]$ is the expected value of $\cdots$ under the sampling distribution $g$. This technique is referred to as "importance sampling" (IS) [34]. The only requirements for the technique to work are:

- we can efficiently sample events $x$ as-per the distribution $g(x)$,

- we can compute $g(x)$ for every sampled event $x$, so that $w$ is exactly computable from (3),

- $f(x) \neq 0 \implies g(x) \neq 0$, i.e., the support of the distribution $g$ contains the support of the distribution $f$.[1]

A key observation at this point is that the quality of the simulated dataset (with regards to a given objective) depends on the sampling distribution $g$. We can then use the freedom of choosing $g(x)$ to match a specific goal. For example, if the distribution $f$ is highly singular, and we want to properly map out the peak and to obtain a reliable estimate of $F$, $g$ needs to match the singularity structure of $f$ well enough to adequately sample the different regions of the phase space.

---

[1]It is sufficient for this condition to be satisfied *almost everywhere*.

More concretely, if $N_s$ simulated events $\{x_1, x_2, \ldots x_{N_s}\}$ are used to estimate[2] $F$ as

$$\hat{F} = \frac{1}{N_s} \sum_{i=1}^{N_s} w(x_i) \, , \tag{5}$$

then the variance of the estimate will be given by

$$\text{var}_g\left[\hat{F}\right] = \frac{1}{N_s^2} \text{var}_g\left[\sum_{i=1}^{N_s} w(x_i)\right] = \frac{\text{var}_g\left[w(x)\right]}{N_s} \, , \tag{6}$$

where $\text{var}_g[\cdots]$ represents the variance of $\cdots$ under the sampling distribution $g$. This means that the quality of the estimation of $F$ will be better if the variance of $w$ is lower. The ideal situation is when $w(x)$ is constant[3] and equal to $F$ for all $x$, or equivalently, if $g(x) = f(x)/F$. In other words, the closer the shape of the sampling distribution $g$ is to the shape of the underlying (unnormalized) theory distribution $f$, the better the estimation of $F$. Instead of minimizing the variance of $w$, an alternative procedure often found in the literature is the minimization of the maximum weight $w_{\text{max}}$, which improves the so called unweighting efficiency $\langle w \rangle / w_{\text{max}}$, whose inverse is the average number of weighted events needed in order to generate one unweighted event distributed as per $f$. This approach also seeks to reduce the distance between the distributions $f$ and $g$, albeit captured by a different measure $w_{\text{max}}$. This has been the guiding principle for previous importance sampling approaches including the VEGAS algorithm [35, 36], Foam [37–40], and more recent machine learning based approaches [41–46]. All these methods implement event sampling strategies that attempt to reduce the distance between the sampling distribution $g$ and the theory distribution $f$.[4]

## 1.2 Optimal Analysis-Specific Importance Sampling (OASIS)

The estimation of $F$ (the cross-section in a HEP setting) is related to the estimation of the expected number of events produced in an experiment from a given process under a given theory model. While this estimation is important, data analysis in HEP has come a long way from the simple counting experiments of the 20th century. In the LHC era, due to the large volume of experimental data available, most analyses study the *distributions* of (possibly multi-dimensional) event variables. The choice of setting $g$ close to $f/F$, as we will discuss, is not optimal for the purposes of these analyses, namely to increase the sensitivity to a) the presence of a signal, or b) the value of a parameter. In fact, minimizing the variance of $w$ is not even optimal for the sensitivity of a simple counting experiment that involves some form of event selection prior to counting. By undersampling the regions of parton-level phase space that tend to fail the event selection cuts, and oversampling the rest of the phase space, one can get a better estimate of the expected number of events passing the event selection criteria using the same number of total simulated events [4]. This, of course, is only one consideration that can influence the choice of the sampling distribution $g$, and can encourage a deviation of $g$ from $f$.

The traditional viewpoint sees importance sampling as just a technique to allow sampling from the distribution $f$, and the weighted nature of the event sample is considered as an associated nuisance to be either controlled (via variance minimization) or efficiently eliminated

---

[2]In what follows, we use a hat symbol to indicate an estimate of some quantity.

[3]This condition needs to only be satisfied *almost everywhere* for optimal estimation of $F$.

[4]An alternative approach, motivated by recent advances in Machine Learning (ML), replaces the exact distribution $f$ with an (approximate) ML regressor trained on a small sample of events produced from $f$ [47–49], leading to an increase in the *rate* but not necessarily the *precision* of the simulation [50]. These methods use points in the phase space for which $f$ is computed, and not events sampled from $f$.

(via unweighting). However, weighted events allow us to not only recycle existing simulated samples for different theory models [51, 52] but also to preferentially focus event sampling in different parts of the phase space [53–55], in line with the goals of the particular analysis at hand. Nature is constrained to produce (unweighted) events as-per the true underlying distribution $f$, whereas weighted simulations of a given theory model are not—in principle, one could use any sampling distribution $g$ satisfying the minimum requirements listed above. With this paper, we hope to kick-start efforts within the HEP community to leverage this freedom in order to produce simulated datasets that offer experimental analyses the opportunity to achieve **better sensitivity per simulated event**, or equivalently, to approach the sensitivity floor (set by other sources of uncertainty) with fewer simulated events. This suggests a unique approach to mitigate the computational resource crunch in HEP. Complementary to ideas that seek to speed up the simulation pipeline, here we propose to reduce the simulation requirements of HEP experiments in the first place, by a) committing to weighted events and b) using a judicious choice of the parton-level phase space sampling distribution $g$. This is very important, because a reduction in the simulation requirements of an experiment represents **a conservation of resources at all stages of the simulation pipeline** including its computational bottleneck, the full detector simulation. We name this approach to importance sampling as Optimal Analysis-Specific Importance Sampling (OASIS). The key ideas of OASIS are the following:

1. Event weighting allows us to sample events according to a distribution $g$ which is *different* from the theory distribution $f$ under consideration.

2. The better sampled a given region of phase space is, the lower the uncertainty on the estimated differential cross-section in that region under the theory distribution $f$.

3. Different regions of phase space are sensitive to different extents to the presence of a signal or to the value of a parameter. Consequently, different regions of phase space offer different levels of utility to the experimental analysis per simulated event, for the same value of $w$ (the ratio between $f$ and the sampling distribution $g$).

4. Adding nuance (and complexity) to the previous point, collider analyses are performed on reconstructed events, typically reduced to low-dimensional event variables, after some event selection and/or categorization. As a result, different parts of the parton-level phase space will be mixed together (probabilistically) before being analyzed, say using histograms or parameterized fits. This means that the quality of inference from events from a particular region of the parton-level phase space depends not just on the sampling distribution in that region, but also on all other regions and other datasets[5] it will be mixed with.

In the rest of the paper, we will use these considerations to derive expressions that quantify the relationship between the sampling distribution $g$ and the sensitivity of the analysis that will use the (weighted) simulated dataset. We will also see how to use these expressions to optimally choose the sampling distribution $g$ that **maximizes the sensitivity of the experiment**, for a given computational budget for simulations—this is the optimality alluded to in the name of the technique.

## 2  OASIS at the parton-level

To warm up, in this section we first tackle the case where the analysis is performed directly on the parton-level event $x$, in its full dimensionality, i.e., without accounting for the complex-

---

[5]The MC datasets for a given analysis could be composed of several background and signal subsamples.

ities mentioned in point 4 above. We will restrict our attention to parameter measurement analyses—note that signal search analyses can be viewed as signal cross-section measurement analyses.

## 2.1 Groundwork

Let $\theta$ be the model parameter being measured in an analysis. The parton-level event distribution $f$, and hence the event weights $w$ will now be parameterized by $\theta$ as indicated by $f(\boldsymbol{x}\,;\,\theta)$ and $w(\boldsymbol{x}\,;\,\theta)$. In this section, we will deal with two kinds of uncertainties:

- **Statistical uncertainties in experimental data:** these are uncertainties originating from the finiteness of the experimental dataset ($N_r < \infty$).

- **Statistical uncertainties in simulated data:** these are uncertainties originating from the finiteness of the simulated datasets ($N_s < \infty$). When reporting experimental results they are usually listed under the broader category of "systematic uncertainties".

The sensitivity of the analysis to the value of $\theta$ near $\theta = \theta_0$ can interpreted as the extent to which values of $\theta$ near $\theta_0$ can be distinguished from each other based on the available data. This is captured by the Fisher information [56] $\mathcal{I}(\theta_0)$ given by

$$\mathcal{I}(\theta_0) = L \int d\boldsymbol{x} \; \frac{1}{f(\boldsymbol{x}\,;\,\theta_0)} \left[ \frac{\partial f(\boldsymbol{x}\,;\,\theta)}{\partial \theta} \right]^2 \bigg|_{\theta=\theta_0} \tag{7a}$$

$$= \int d\boldsymbol{x} \; \frac{1}{L f(\boldsymbol{x}\,;\,\theta_0)} \left[ L \frac{\partial f(\boldsymbol{x}\,;\,\theta)}{\partial \theta} \right]^2 \bigg|_{\theta=\theta_0} , \tag{7b}$$

where $L$ is the integrated luminosity of the experiment. Note that this is the Fisher information contained in the entire dataset treated as a random variable, and not just a single event, hence the presence of $L$ in the expression. We derive this expression for the Fisher information for the case when the total number of events is a Poisson distributed random variable (with $\theta$ dependent mean) in Appendix A. If the true value of the parameter is $\theta_0$, $[\mathcal{I}(\theta_0)]^{-1}$ sets the lower limit on the variance of any unbiased estimator $\hat{\theta}$ for $\theta$ constructed out of the experimental data, according to the Cramér–Rao bound [56]. Furthermore, this lower bound is achieved in the asymptotic limit by the Maximum Likelihood Estimator (MLE), provided the estimation is performed using the exact functional form for the true distribution $f$ [57]. In our case, however, this bound cannot be achieved since we do not know the exact functional form of $f$ and are only using a MC estimate for it.

The expression for the Fisher information accounts only for the statistical uncertainties in experimental data. However, here we want to additionally account for the fact that realistic analyses rely on finite simulations to perform the parameter estimation—in particular $N_s$ events sampled as per a given sampling distribution $g(\boldsymbol{x})$ and weighted accordingly. To incorporate the uncertainty due to the finiteness of the simulated sample, let us first intuitively understand the expression for the Fisher information in (7b). Let $n_r$ be the number of events in a small bin of size $\Delta\boldsymbol{x}$ at a given value of $\boldsymbol{x}$, for a given value of $\theta = \theta_0$. Since $n_r$ is a Poisson distributed random variable, its mean and variance are both given by

$$E_f[n_r] = \mathrm{var}_f[n_r] = L f(\boldsymbol{x}\,;\,\theta_0)\Delta\boldsymbol{x} , \tag{8}$$

and the corresponding standard deviation is

$$\mathrm{stdev}_f[n_r] = \sqrt{L f(\boldsymbol{x}\,;\,\theta_0)\Delta\boldsymbol{x}} . \tag{9}$$

The difference in the expected counts under two different values of $\theta$ near $\theta_0$ that differ by a small value $\delta\theta$ is given by:

$$\delta E_f[n_r] \equiv L \, \delta f \, \Delta\boldsymbol{x} = \left[ L \, \frac{\partial f(\boldsymbol{x}\,;\,\theta)}{\partial\theta} \right]\Bigg|_{\theta=\theta_0} \Delta\boldsymbol{x}\,\delta\theta \,. \tag{10}$$

The statistical significance of this difference depends on its relative size with respect to the standard deviation of $n_r$ given by (9). Now, (7b) can be seen as an analogue of the familiar "(deviation over standard deviation) summed over bins in quadrature" per unit $(\delta\theta)^2$. This line of reasoning lets us see why a higher value of $\mathcal{I}(\theta_0)$ corresponds to a greater distinguishability between neighboring values of $\theta$.

Armed with this intuitive understanding, we can now introduce the statistical uncertainty from the simulated data into the expression for the Fisher information. Let the random variable $n_s$ be the number of simulated events in a small bin of size $\Delta\boldsymbol{x}$ at a given value of $\boldsymbol{x}$, for a given value of $\theta=\theta_0$. The mean and variance of $n_s$ are both given by

$$E_g[n_s] = \text{var}_g[n_s] = N_s\,g(\boldsymbol{x})\,\Delta\boldsymbol{x} \,. \tag{11}$$

Recall that the main purpose of doing the MC simulations was to construct an estimate for $E_f[n_r]$, say $n_{\text{est}}$, out of $n_s$. This can be done by scaling $n_s$ by appropriate factors to account for a) the actual integrated luminosity in the experiment and b) the difference between the sampling distribution $g$ and the true distribution $f$:

$$n_{\text{est}} \equiv n_s \times \frac{L}{N_s} \times w(\boldsymbol{x}\,;\,\theta_0) \,. \tag{12}$$

The expected value of this estimate $n_{\text{est}}$ under $g$ (for a given value of $\boldsymbol{x}$) equals $E_f[n_r]$ as it should:

$$E_g\left[n_{\text{est}}\right] = E_g[n_s] \times \frac{L\,w(\boldsymbol{x}\,;\,\theta_0)}{N_s} \tag{13a}$$

$$= N_s\,g(\boldsymbol{x})\,\Delta\boldsymbol{x} \times \frac{L\,w(\boldsymbol{x}\,;\,\theta_0)}{N_s} = L\,f(\boldsymbol{x}\,;\,\theta_0)\,\Delta\boldsymbol{x} = E_f[n_r] \,, \tag{13b}$$

where we have used (3), (8), and (11). The variance of the estimate $n_{\text{est}}$ is given by:

$$\text{var}_g\left[n_{\text{est}}\right] = \text{var}_g\left[n_s\right] \times \left[\frac{L\,w(\boldsymbol{x}\,;\,\theta_0)}{N_s}\right]^2 \tag{14a}$$

$$= N_s\,g(\boldsymbol{x})\,\Delta\boldsymbol{x}\,\frac{L^2\,w^2(\boldsymbol{x}\,;\,\theta_0)}{N_s^2} \tag{14b}$$

$$= \frac{L}{N_s}\,w(\boldsymbol{x}\,;\,\theta_0) \times L\,f(\boldsymbol{x}\,;\,\theta_0)\,\Delta\boldsymbol{x} \,. \tag{14c}$$

Now, using (8) and (14c), we can add the statistical uncertainty from the real data to the statistical uncertainty from the simulated data in quadrature[6] as

$$\text{var}_f[n_r] + \text{var}_g[n_{\text{est}}] = L\,f(\boldsymbol{x}\,;\,\theta_0)\left[1 + \frac{L}{N_s}\,w(\boldsymbol{x}\,;\,\theta_0)\right]\Delta\boldsymbol{x} \,, \tag{15}$$

which allows us to modify the expression for the Fisher information in (7b) as

$$\mathcal{I}_{\text{MC}}(\theta_0) = \int d\boldsymbol{x}\,\frac{\left[L\,\dfrac{\partial f(\boldsymbol{x}\,;\,\theta)}{\partial\theta}\right]^2\Bigg|_{\theta=\theta_0}}{L\,f(\boldsymbol{x}\,;\,\theta_0)\left[1 + \dfrac{L}{N_s}\,w(\boldsymbol{x}\,;\,\theta_0)\right]} \,, \tag{16}$$

---

[6]A subtle point is that these are not the uncertainties *estimated* from the real or simulated datasets, but rather the uncertainties *expected* in them under the true and sampling distributions $f$ and $g$.

where we have replaced the denominator in (7b) with the corresponding expression from (15) and the subscript "MC" indicates that this version of the Fisher information incorporates the uncertainty from the MC simulation [2]. By construction, $\mathcal{I}_{\text{MC}}$ captures the sensitivity of the experiment to the value of a parameter, when the analysis is performed by comparing the real dataset against simulations (likelihood-free inference).

Since $\mathcal{I}_{\text{MC}}$ scales linearly with the integrated luminosity (for a given value of $L/N_s$), we can factor it out and rewrite (16) as

$$\Rightarrow \frac{\mathcal{I}_{\text{MC}}(\theta_0)}{L} = \int d\boldsymbol{x}\; f(\boldsymbol{x}\,;\,\theta_0)\; \frac{\left[\partial_\theta \ln[f(\boldsymbol{x}\,;\,\theta)]\right]^2\Big|_{\theta=\theta_0}}{1 + \dfrac{L}{N_s} w(\boldsymbol{x}\,;\,\theta_0)} \tag{17a}$$

$$= \int d\boldsymbol{x}\; f(\boldsymbol{x}\,;\,\theta_0)\; \frac{u^2(\boldsymbol{x}\,;\,\theta_0)}{1 + \dfrac{L}{N_s} w(\boldsymbol{x}\,;\,\theta_0)}\;, \tag{17b}$$

where $u(\boldsymbol{x}\,;\,\theta)$ is defined as

$$u(\boldsymbol{x}\,;\,\theta) \equiv \partial_\theta \ln[f(\boldsymbol{x}\,;\,\theta)] = \frac{1}{f(\boldsymbol{x}\,;\,\theta)} \frac{\partial f(\boldsymbol{x}\,;\,\theta)}{\partial \theta}\;. \tag{18}$$

We will refer to $u(\boldsymbol{x}\,;\,\theta)$ as the *per-event score* at the parton-level. This is related to the *score* of an observation used in the statistics literature, with the only distinction being that the integral $F$ of the distribution $f$ over the phase space is $\theta$ dependent, while the traditional score uses a normalized probability distribution[7]. The per-event score in (17b) captures the sensitivity of (the weight) of a given event to the value of $\theta$. It can be computed directly from the parton-level oracle used to compute $f(\boldsymbol{x}\,;\,\theta)$.

The quantity $\mathcal{I}_{\text{MC}}(\theta_0)$ captures the relationship between the sampling distribution $g$ and the sensitivity of the analysis. Correspondingly, $\mathcal{I}_{\text{MC}}(\theta_0)$ can be used as a performance measure to be *maximized* by the *optimal* sampling distribution. Note that $g(\boldsymbol{x})$ features in (17b) only through the weight $w(\boldsymbol{x}) = f(\boldsymbol{x})/g(\boldsymbol{x})$, so that for a given $\boldsymbol{x}$, lower values of $w$ correspond to higher sampling rates $g$. This is consistent with the integrand in (17b) being negatively correlated with the weight $w$, since the integrand should increase with increasing sampling rate. However, we cannot assign small weights for all regions since, according to (4), the weights $w(\boldsymbol{x})$ are constrained to have an expected value of $F$ under the sampling distribution $g$. In other words, we are playing a "fixed sum game"—increasing the sampling rate in some regions must be accompanied by a corresponding decrease in the sampling rate in others. As a result, the sampling of different regions will need to be prioritized based on the true distribution $f$ and the per-event score $u$ of the events in the different regions—this is what we set out to do using OASIS. Before discussing how to use (17b) to construct a good sampling distribution $g$, let us consider some special cases in order to gain some further intuition.

## 2.2 Special cases of $\mathcal{I}_{\text{MC}}$

In this subsection, we shall discuss several special cases of the main result (17b). For notational convenience, from now on the $\theta_0$ dependence of $f$, $F$, $w$, $u$, $\mathcal{I}_{\text{MC}}$, $\mathcal{I}$, the optimal sampling distribution, etc., will not be explicitly indicated unless deemed useful.

---

[7]$\partial_\theta \ln(f/F)$ (as used in [58–60]) naturally appears in the expression for Fisher information when the experiment uses a given fixed number of real events, while $\partial_\theta \ln f$ naturally appears when the experiment has a given fixed integrated luminosity (with the actual number of events being a Poisson distributed random variable). The distinction should not matter much as long as $F$ is only weakly dependent on $\theta$.

### 2.2.1 Sampling directly from the true distribution

In order to make the connection to the conventional use of importance sampling, let us first consider the case when $g$ mimics $f$ well. For concreteness, let us assume that $g$ matches the true distribution $f$ exactly, i.e.

$$g(\boldsymbol{x}) = \frac{f(\boldsymbol{x})}{F} \ . \tag{19}$$

As a result, the weights are constant, $w = F$, and the denominator of the integrand in (17) can be taken out in front of the integral:

$$\mathcal{I}_{\text{MC}} = \frac{1}{1 + \dfrac{L\,F}{N_s}}\, I \approx \frac{1}{1 + \dfrac{N_r}{N_s}}\, I \ . \tag{20}$$

Note that $L\,F$ is the *expected* total number of events in the real dataset—barring some spectacular surprise in the data, this estimate should not be too far off from the number of events $N_r$ which were actually observed, hence the last approximate relation above. The prefactor $(1 + LF/N_s)^{-1}$ is precisely the additional penalty that we have to incur for using finite simulation samples. Thus (20) quantifies the dependence of the sensitivity on the size of the simulated dataset $N_s$—the greater the value of $N_s$, the greater the sensitivity, but the returns diminish as the value of $N_s$ gets too large, due to the additive term "1" in the denominator.

### 2.2.2 Constant per-event score

As a second example, consider the case when the per-event score $u(\boldsymbol{x})$ is constant in $\boldsymbol{x}$:

$$u(\boldsymbol{x}) = \text{const.} \tag{21}$$

Then (17b) gives

$$\frac{\mathcal{I}_{\text{MC}}}{L} = u^2 \int d\boldsymbol{x}\ g(\boldsymbol{x})\,\frac{w(\boldsymbol{x})}{1 + \dfrac{L}{N_s}\,w(\boldsymbol{x})} \tag{22a}$$

$$\leq u^2\,\frac{E_g[w(\boldsymbol{x})]}{1 + \dfrac{L}{N_s}\,E_g[w(\boldsymbol{x})]} \tag{22b}$$

$$= u^2\,\frac{F}{1 + \dfrac{LF}{N_s}} \approx \frac{u^2}{L}\,\frac{N_r}{1 + \dfrac{N_r}{N_s}}\ , \tag{22c}$$

where $E_g[\cdots]$, as before, represents the expectation value under the sampling distribution $g$. In (22b), we have used Jensen's inequality [56,61] and the fact that $w/(1 + \alpha w)$ is a concave function in $w$ for a positive $\alpha$. The equality in (22b) holds iff $w(\boldsymbol{x}\,;\theta_0)$ is a constant almost everywhere. In other words, when the per-event score is immaterial, $\mathcal{I}_{\text{MC}}$ is maximized if $g(\boldsymbol{x}) = f(\boldsymbol{x})/F$ almost everywhere, which is in agreement with conventional wisdom.

This special case suggests that in the general case when the per-event score is not a constant, the optimal sampling distribution $g_{\text{optimal}} \equiv f/w_{\text{optimal}}$ will be such that the weights $w_{\text{optimal}}$ depend only on the (absolute value of the) per-event score. This hunch will play a crucial role in constructing $g_{\text{optimal}}$ later on in Section 2.3.3.

### 2.2.3 Insufficient simulated data

Let us now consider the (unfortunate) situation when the amount of simulated data is very limited, i.e.,

$$N_s \, g(\boldsymbol{x}) \ll L \, f(\boldsymbol{x}), \tag{23}$$

implying that in every region of the phase space the simulated data is much sparser than the real dataset. In this case (17b) gives

$$\frac{\mathcal{I}_{\mathrm{MC}}}{L} \approx \int d\boldsymbol{x} \; f(\boldsymbol{x}) \; \frac{u^2(\boldsymbol{x})}{\dfrac{L}{N_s} \, w(\boldsymbol{x})} \tag{24}$$

$$\Rightarrow \mathcal{I}_{\mathrm{MC}} \approx N_s \int d\boldsymbol{x} \; g(\boldsymbol{x}) \, u^2(\boldsymbol{x}) \tag{25}$$

and $\mathcal{I}_{\mathrm{MC}}$ is maximized when the sampling distribution $g$ focuses entirely on the most sensitive regions, i.e., those with the highest magnitude of per-event score $|u(\boldsymbol{x})|$, since they offer the best bang for the buck in terms of sensitivity gained per event generated.

The additive term "1" in the denominator in (17b) was neglected in the limit (23) to arrive at (24), but its role is to capture the diminishing returns associated with an indefinite increase of the simulated statistics in some region (or increasing the total number of simulated event $N_s$) as we approach the uncertainty floor set by the statistical uncertainties in the real dataset. This term will eventually force the optimal sampling distribution to also cover other regions of the phase space with lower values of $|u(\boldsymbol{x})|$, albeit with higher weights.

### 2.2.4 Too much simulated data

Finally, let us consider the opposite limit to (23), i.e., when the simulated data is much denser than the real dataset (in every region of the phase space)

$$N_s \, g(\boldsymbol{x}) \gg L \, f(\boldsymbol{x}) \,. \tag{26}$$

Expanding the denominator in (17b), one obtains

$$\frac{\mathcal{I}_{\mathrm{MC}}}{L} \approx \int d\boldsymbol{x} \; f(\boldsymbol{x}) \left[ 1 - \frac{L}{N_s} \, w(\boldsymbol{x}) \right] u^2(\boldsymbol{x}) \tag{27a}$$

$$= \frac{I}{L} - \frac{L}{N_s} \int d\boldsymbol{x} \; g(\boldsymbol{x}) \, w^2(\boldsymbol{x}) \, u^2(\boldsymbol{x}) \,. \tag{27b}$$

In Appendix B we show that this expression is maximized when $g(\boldsymbol{x}) \propto f(\boldsymbol{x}) |u(\boldsymbol{x})|$, i.e.,

$$\lim_{N_s \to \infty} g_{\mathrm{optimal}}(\boldsymbol{x}) = \frac{f(\boldsymbol{x}) \, |u(\boldsymbol{x})|}{\displaystyle \int d\boldsymbol{x} \; f(\boldsymbol{x}) |u(\boldsymbol{x})|} \,. \tag{28}$$

The corresponding optimal weights (in the large $N_s$ limit) are proportional to $|u(\boldsymbol{x})|^{-1}$.

### 2.3 Constructing optimal sampling distributions

In this subsection, we will introduce some prescriptions for constructing the optimal sampling distribution that maximizes $\mathcal{I}_{\mathrm{MC}}$. We will refer to this procedure as "training the sampling distribution" regardless of whether machine learning techniques are used or not. Instead of reinventing the wheel, we will piggyback on existing importance sampling techniques whenever possible. Equation (17b) will serve as the starting point for all our prescriptions.

We will assume that we are provided an oracle that can be queried for the value of $f(\boldsymbol{x}\,;\,\theta_0)$ and $u(\boldsymbol{x}\,;\,\theta_0)$ for different events $\boldsymbol{x}$. $\theta_0$ is the value of the parameter $\theta$ at which the dataset is to be produced, and it is also the parameter value near which we want the simulated dataset to offer the most sensitivity. In situations where the same simulated dataset will be reweighted for different values of $\theta$ [51], a representative value of $\theta$ can be used as $\theta_0$ for the purpose of training the sampling distribution.

The term $L/N_s$ in (17b) is a predetermined[8] parameter that specifies the size of he simulated dataset that will be generated using the OASIS-trained sampling distribution. Note that the expected number of events in the real dataset is $LF$. So, $L/N_s$ can be thought of as the ratio $N_r/N_s$ of real to simulated events used by the analysis, up to a factor of $1/F$ which can be estimated using a preliminary or preexisting dataset. As we will see in Figure 6 and Figure 7, sampling distributions trained at some value of $L/N_s$ will continue to be good at other values as well—in this sense, $L/N_s$ is just a heuristic parameter and an accurate pre-estimation of the parameter is not critical to the utility of the trained sampling distribution.

### 2.3.1 Adjusting the weights of cells in phase space

Foam [37–40] is an importance sampling technique under which the phase space of the events is divided into several non-overlapping cells, with each cell $i$ having an associated probability, say $p_{\text{cell}\,i}$, and an associated phase space volume $V_{\text{cell}\,i}$. The individual cell-probabilities sum up to 1:

$$\sum_i p_{\text{cell}\,i} = 1 \,, \tag{29}$$

and the individual cell volumes add up to the total phase space volume $V_{\text{tot}}$:

$$\sum_i V_{\text{cell}\,i} = V_{\text{tot}} \,. \tag{30}$$

After constructing the cells (and their associated probabilities), the sampling of an event under Foam is done is two steps: 1) choose a cell as per the probabilities $p_{\text{cell}}$, and 2) choose an event uniformly at random within the chosen cell. This "piecewise uniform" sampling distribution is given by

$$g(\boldsymbol{x}) = \frac{p_{\text{cell}(\boldsymbol{x})}}{V_{\text{cell}(\boldsymbol{x})}} \,, \tag{31}$$

where $\text{cell}(\boldsymbol{x})$ is the cell that event $\boldsymbol{x}$ belongs to. A simple approach to performing OASIS is to work with the cells induced by Foam (or a different cellular importance sampling technique[9]), and simply adjust the probabilities $p_{\text{cell}}$ for choosing the different cells.[10]

For the piecewise uniform sampling distribution given in (31), the expression for $\mathcal{I}_{\text{MC}}/L$ given in (17b) can be written as

$$\frac{\mathcal{I}_{\text{MC}}}{L} = \int d\boldsymbol{x}\, f(\boldsymbol{x}) \, \frac{u^2(\boldsymbol{x})}{1 + \dfrac{L}{N_s} \dfrac{f(\boldsymbol{x})\, V_{\text{cell}(\boldsymbol{x})}}{p_{\text{cell}(\boldsymbol{x})}}} \,. \tag{32}$$

Since we do not a priori know the functional form of $f$, training the sampling distribution $g$ needs to be performed using simulated events, possibly sampled from a different sampling

---

[8]The integrated luminosity $L$ is fixed from experiment while the total number of simulated events $N_s$ depends on the available computing resource budget.

[9]Since VEGAS assumes the factorizability of the integrand, it may not offer sufficient flexibility to adjust the probabilities of the individual cells of the rectangular grid.

[10]Although the cells in Foam were not originally constructed for the purpose of OASIS, they will still capture the singular features shared by the optimal sampling distribution $g(\boldsymbol{x})$ and the true distribution $f(\boldsymbol{x}\,;\,\theta_0)$.

distribution, say $g'$ (with weights $w' = f/g'$) [11]. This lets us rewrite the previous equation as

$$\frac{\mathcal{I}_{\text{MC}}}{L} = \int d\boldsymbol{x} \; g'(\boldsymbol{x}) \, w'(\boldsymbol{x}) \, \frac{u^2(\boldsymbol{x})}{1 + \dfrac{L}{N_s} \dfrac{g'(\boldsymbol{x}) \, w'(\boldsymbol{x}) \, V_{\text{cell}(\boldsymbol{x})}}{p_{\text{cell}(\boldsymbol{x})}}} \; . \tag{33}$$

A further (optional) simplification of the expression is possible if the distribution $g'$ is also piecewise uniform with the exact same cells used by $g$ (as will be the case when data generated as-per the "regular" importance sampling `Foam` is used to adjust its cell-probabilities). If $g'(\boldsymbol{x}) = p'_{\text{cell}(\boldsymbol{x})}/V_{\text{cell}(\boldsymbol{x})}$, then

$$\frac{\mathcal{I}_{\text{MC}}}{L} = \int d\boldsymbol{x} \; g'(\boldsymbol{x}) \, w'(\boldsymbol{x}) \, \frac{u^2(\boldsymbol{x})}{1 + \dfrac{L}{N_s} \dfrac{p'_{\text{cell}(\boldsymbol{x})} w'(\boldsymbol{x})}{p_{\text{cell}(\boldsymbol{x})}}} \; . \tag{34}$$

Now, if the cell-probabilities $p_{\text{cell}(\boldsymbol{x})}$ are parameterized by parameters $\boldsymbol{\varphi}$, then the gradient of $\mathcal{I}_{\text{MC}}$ with respect to $\boldsymbol{\varphi}$ can be written as

$$\frac{N_s}{L^2} \nabla_{\boldsymbol{\varphi}} \mathcal{I}_{\text{MC}} = \int d\boldsymbol{x} \; g'(\boldsymbol{x}) \left[ \frac{p'_{\text{cell}(\boldsymbol{x})} w'^2(\boldsymbol{x}) \, u^2(\boldsymbol{x})}{\left[ p_{\text{cell}(\boldsymbol{x})} + \dfrac{L}{N_s} p'_{\text{cell}(\boldsymbol{x})} w'(\boldsymbol{x}) \right]^2} \left[ \nabla_{\boldsymbol{\varphi}} \, p_{\text{cell}(\boldsymbol{x})} \right] \right] \; . \tag{35}$$

This expression facilitates the usage of stochastic or (mini)-batch gradient **ascent** to find the optimal parameters $\boldsymbol{\varphi}$ that **maximize** $\mathcal{I}_{\text{MC}}$. The discrete cell-probabilities (non-negative and summing up to 1) can be parameterized with real valued $\boldsymbol{\varphi}$ (with the same dimensionality as the number of cells), using the softmax function [62] $\boldsymbol{\sigma}$ as

$$p_{\text{cell } i} = \sigma_i(\boldsymbol{\varphi}) \equiv \frac{e^{\varphi_i}}{\sum_j e^{\varphi_j}} \; . \tag{36}$$

Under this parameterization, the gradient of $p_{\text{cell}(\boldsymbol{x})}$ with respect to $\boldsymbol{\varphi}$ in (35) can be computed using

$$\frac{\partial p_{\text{cell } i}}{\partial \varphi_j} = p_{\text{cell } i} \left( \delta_{ij} - p_{\text{cell } j} \right) , \tag{37}$$

where $\delta_{ij}$ is the Kronecker delta function.

### 2.3.2 An illustrative example

Now we will present a simple one-dimensional example that demonstrates

A. How the technique introduced in Section 2.3.1 can be used to train the sampling distribution.

B. How an OASIS-trained sampling distribution $g$ can offer more sensitivity to the experiment than even the ideal[12] case of regular importance sampling (IS).

---

[11]The events which were generated during the construction of the cells could also be reused for the purpose of this cell-probability adjustment. But one will have to keep track of the sampling distributions used to generate points at different stages in order to reuse them later.

[12]The ideal case of IS is when the sampling distribution perfectly matches the normalized true distribution $f/F$. In order to be conservative in our evaluation of the performance of regular importance sampling, from now on any mention of IS will refer to this ideal situation.

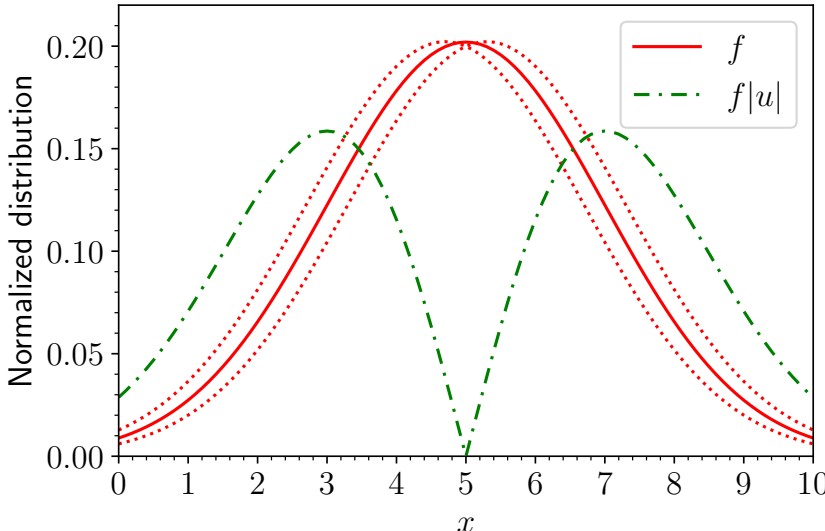

Figure 1: The red lines show distributions $f(x)$ given by (38) for $\theta = \theta_0 = 5$ (solid line) and $\theta = \theta_0 \pm 0.3$ (dotted lines). The green dot-dashed line shows the distribution $f(x)|u(x)|$ for $\theta = \theta_0$. All distributions have been unit-normalized in the range $0 \leq x \leq 10$.

Despite this being only a one-dimensional toy example, it will be sufficient to convey our main point. Later in the paper we will additionally show

- How to reduce the task of training the sampling distributions in higher dimensions to a one-dimensional problem. This means that the ability to derive the optimal sampling distribution for a one-dimensional case is sufficient to perform OASIS in higher dimensions as well.

- How the results from our one-dimensional example correlate with existing plots from a realistic physics analysis by the Compact Muon Solenoid (CMS) experiment, thus confirming the potential for resource conservation offered by OASIS to collider experiments.

The toy example we will consider is the measurement of the mean of a normally distributed random variable of known standard deviation, which we take to be equal to 2:

$$f(x\,;\,\theta) = \frac{1}{\sqrt{8\pi}}\,\exp\left[-\frac{(x-\theta)^2}{8}\right]. \tag{38}$$

The per-event score for this distribution is given by

$$u(\boldsymbol{x}\,;\,\theta) = \frac{x-\theta}{4}\,. \tag{39}$$

In a real analysis, these functional forms will be unavailable, and the analysis will be performed using only the simulated datasets, without knowledge of the true model or the fact that the parameter $\theta$ is simply the mean of the distribution (likelihood-free inference).

We will fix the phase space region considered by the analysis to be $0 \leq x \leq 10$, and we will choose $\theta_0 = 5$. The red solid curve in Figure 1 shows the distribution $f(x\,;\,\theta_0)$, while the red dotted curves show the same distribution $f$, but for neighboring values of $\theta = \theta_0 \pm \epsilon$ with $\epsilon = 0.3$. By visually inspecting these three curves, one can get a feeling for the sensitivity to the parameter $\theta$ at different values of $x$. Note that the region near the maximum of the distribution

Table 1: The parameters used in the mini-batch gradient ascent.

| $\frac{L}{N_s}$ | Initial $\varphi$ | Number of steps | Learning rate $\eta$ | Mini-batch size |
|---|---|---|---|---|
| 1 | $(1, 1, \ldots, 1)$ | $50,000$ | $0.1$ | $20$ |
| 10 | $(1, 1, \ldots, 1)$ | $50,000$ | $0.1$ | $20$ |
| 0.1 | $(1, 1, \ldots, 1)$ | $100,000$ | $0.5$ | $20$ |

offers *the least* amount of sensitivity, since the value of $f$ there does not change much as we vary $\theta$. On the other hand, the regions away from the peak appear to be much more sensitive to $\theta$, the exact amount being a function of the per-event score $u$ and the accumulated statistics as encoded by the value of $f$. In the large simulation statistics limit, the optimal sampling distribution from (28) is proportional to the product $f|u|$ (at $\theta = \theta_0$), which is shown in Figure 1 with the green dot-dashed line. The shape of the $f|u|$ distribution is such that its maxima are located one standard deviation from $x = \theta_0$, which can be checked explicitly using the exact expressions (38) and (39). We see that going after a sampling distribution $g$ which resembles the red lines in Figure 1 (as in the standard approaches to IS) and thus tries to populate the region of the peak, is sub-optimal and this sub-optimality is what OASIS sets out to fix.

In order to train the sampling distribution using (35), we split the region $0 \le x \le 10$ into 20 bins of equal length 0.5. For simplicity $p'_{\text{cell}}$ was set to be equal for all cells ($p'_{\text{cell}} = 1/20$). This induces a uniform distribution $g'(x) = 1/10$ for the sampling of the training events. For this first exercise, we set the heuristic parameter $L/N_s$ to 1. Since the integral of $f(x)$ in the region $[0, 10]$ is approximately 1 ($\approx 0.9876$), this choice of heuristic parameter corresponds to roughly equal number of simulated and real events[13].

Using 10,000 training events sampled as-per $g'$, the cell-probabilities $p_{\text{cell}}$ were trained using mini-batch gradient ascent, i.e., using a mini-batch of randomly chosen training events in each training step. The operation performed in each step is given by

$$
\varphi \to \varphi + \eta \times \underset{\text{mini-batch}}{\text{AVG}} \left[ \frac{p'_{\text{cell}(x)} \, w'^{\,2}(x) \, u^2(x)}{\left[ p_{\text{cell}(x)} + \frac{L}{N_s} p'_{\text{cell}(x)} w'(x) \right]^2} \left[ \nabla_\varphi \, p_{\text{cell}(x)} \right] \right] , \tag{40}
$$

where $\eta$ is the learning rate parameter, and AVG is the average over the mini-batch. The first row of Table 1 summarizes the parameters of the mini-batch optimization for $L/N_s = 1$. The OASIS-trained sampling distribution $g^*(x)$ given by

$$
g^*(x) = \frac{p^*_{\text{cell}(x)}}{V_{\text{cell}(x)}} , \tag{41}
$$

where $p^*_{\text{cell}}$ refers to the trained values of cell-probabilities, is shown in the left panel of Figure 2 (the blue dashed curve). For comparison, the true distribution $f(x; \theta_0)$, which is the ideal case in regular importance sampling (IS), is shown with the red solid line, after normalizing to 1 in the analysis region $0 \le x \le 10$. Since the OASIS sampling distribution (41) is different from the true distribution, each sampled event has an associated weight. The magenta curve in the right panel of Figure 2 shows the ratio of weights for events sampled under

---

[13]In our toy example, the cross-section $F$, and hence, the integrated luminosity $L$ are both dimensionless quantities. Although different, this is compatible with the HEP convention where $L$ and $F$ are both dimensionful, but their product is dimensionless.

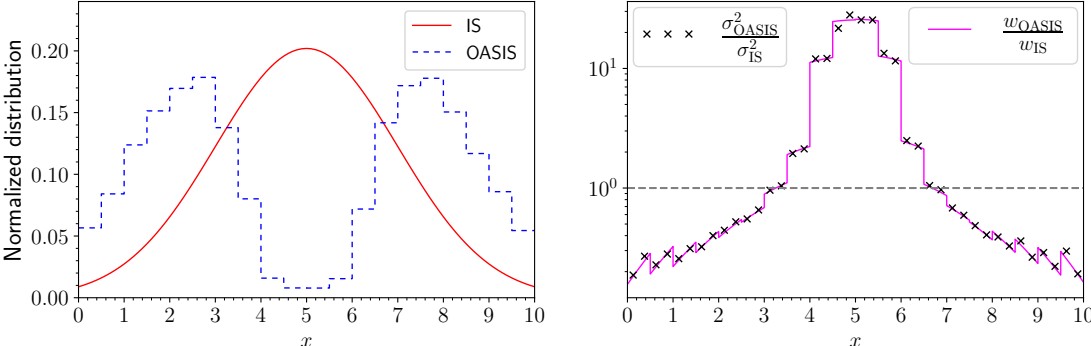

Figure 2: The blue dashed line in the left panel shows the sampling distribution trained using OASIS with $L/N_s$ set to 1. For comparison, the true distribution $f(x; \theta_0)$, which is the ideal case in regular importance sampling (IS), is shown with the red solid line. The magenta line in the right panel shows the ratio of weights for events sampled under the OASIS-trained distribution (41) versus events sampled under the best-case scenario in regular IS (i.e., $g$ matching $f$ perfectly). The $\times$ marks indicate the ratios of the squares of the estimated error-bars within each bin for the histograms in Figure 3 produced under OASIS and IS.

the OASIS-trained distribution (41) versus events sampled under the best-case scenario in regular IS when $g$ matches $f/F$ perfectly (henceforth referred to as the IS distribution). Note that all the events from the IS-generated distribution will have *constant* weights $w_{IS}$ equal to $F(\theta_0)$ (which is $\approx 0.9876$). Notice how the regions oversampled (undersampled) under OASIS have $w_{OASIS}/w_{IS}$ less than 1 (greater than 1). Also note how the discontinuities in the sampling distribution coincide with the discontinuities in the weight function. The edge effects from these two discontinuities at the cell/bin boundaries will cancel out and the weighted dataset will not have any binning artifacts—this is true of existing cellular importance sampling techniques as well.

Next we generate 100,000 events from the OASIS-trained sampling distribution (41), weight them appropriately, and plot a normalized histogram in the top panel of Figure 3 (blue histogram). For comparison we also show a histogram with 100,000 events from the IS-sampled distribution (red histogram). To allow for visual comparison, the histograms are plotted on a log-scale on two different vertical axes, which are slightly displaced, so that the IS (OASIS) values should be read off from the red $y$-axis on the left (the blue $y$-axis on the right). The bottom panel shows the ratio of the simulated counts to the true expected count (calculated by integrating the true distribution within each bin). As expected, the histograms from the IS and from the OASIS-trained sampling distributions are both consistent with the true distribution, owing to the robustness of importance sampling as a Monte Carlo technique.

Notice that near the center of the histogram ($x = 5$) the OASIS-trained histogram in Figure 3 has larger error-bars than the IS histogram. Away from the center, however, the situation is reversed, with OASIS leading to smaller error-bars than the IS distribution. This is because of the preferential sampling performed by the OASIS-trained sampling distribution $g^*$: we already saw from the left panel in Figure 2 that the maximum sensitivity is expected away from the peak. Correspondingly, the OASIS-trained sampling distribution $g^*$ is designed so that the resulting OASIS-trained histogram has relatively small error bars in the $\theta$-sensitive regions of phase space, $x \in (0, 3.5)$ and $x \in (6.5, 10)$. This benefit, however, comes at the cost or allowing larger error bars elsewhere, in this case in $x \in (3.5, 6.5)$, but that is precisely where the sensitivity to $\theta$ is minimal, and such regions are anyway not very useful to an experimental analysis which is trying to measure $\theta$. In summary, as illustrated in Figure 3, the

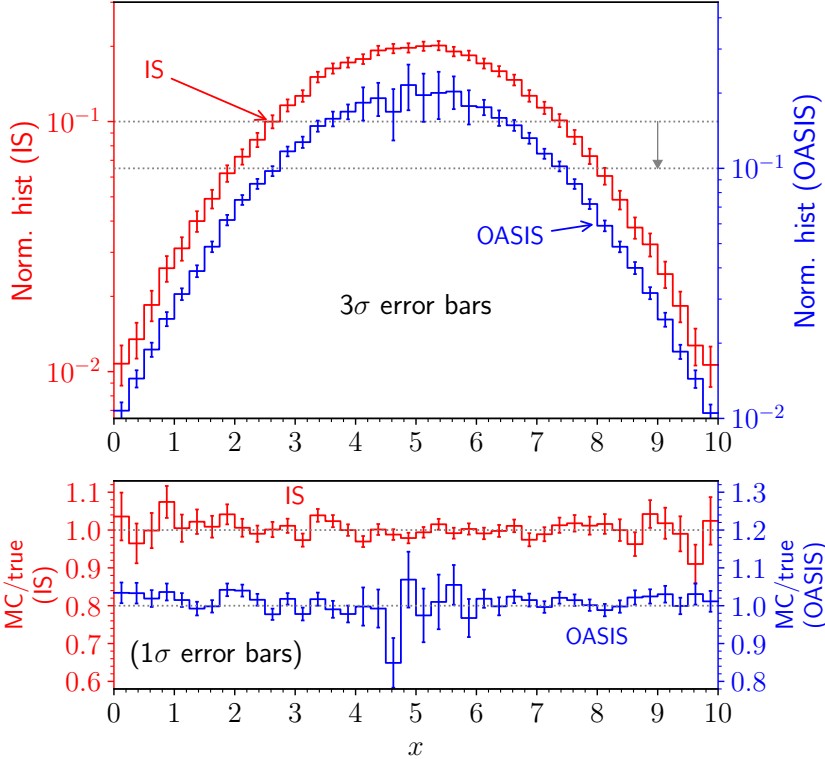

Figure 3: Normalized histogram counts with $3\sigma$ error bars with 100,000 events sampled (and weighted appropriately) under the OASIS-trained distribution (blue lines) and the true distribution (red lines labelled IS). To allow for visual comparison, the histograms are plotted on a log-scale and their corresponding $y$-axes have been shifted vertically as indicated by the vertical arrow. The bottom panel shows the ratio of the simulated counts to the true expected count (calculated by integrating the true distribution within each bin), along with $1\sigma$ error bars.

OASIS perspective keeps an eye on the big picture and improves the precision of the simulated data precisely in the regions that are most valuable to the experimental analysis which will be making use of that simulated data later on.

The ratios of squares of the estimated error-bars of the bin counts of the histograms produced under OASIS and IS in Figure 3 are shown in the right panel of Figure 2 with × marks. They match the weight ratios (magenta curve) since, for a given bin, the statistical error $\sigma$ in the simulated (normalized) histogram scales as $\sigma \propto 1/\sqrt{g} \propto \sqrt{w}$. The slight mismatch between the × marks and the magenta curve is present because we use the uncertainties *estimated* from the finite simulated samples in Figure 3—it is the uncertainties in the uncertainty estimates themselves which cause the mismatch.

Next we repeat the exercise and train optimal sampling distributions for different choices of $L/N_s$ (with optimization parameters shown in Table 1). The trained distributions are shown in the upper panel of Figure 4: $L/N_s = 10$ (red solid line), $L/N_s = 1$ (blue dashed line), $L/N_s = 0.1$ (magenta dotted line). The green dot-dashed line shows the theoretical optimal sampling distribution in the large $N_s$ limit from (28) (it is the same green dot-dashed line appearing in Figure 1). The per-event score $|u|$ is plotted in the lower panel of Figure 4. Note how for large values of $L/N_s$, the optimal sampling distribution $g^*$ aggressively focuses the event sampling in regions of high $|u|$, while at lower values of $L/N_s$, the sampling distribution is more lenient towards regions of lower $|u|$. This trend is expected from the special cases

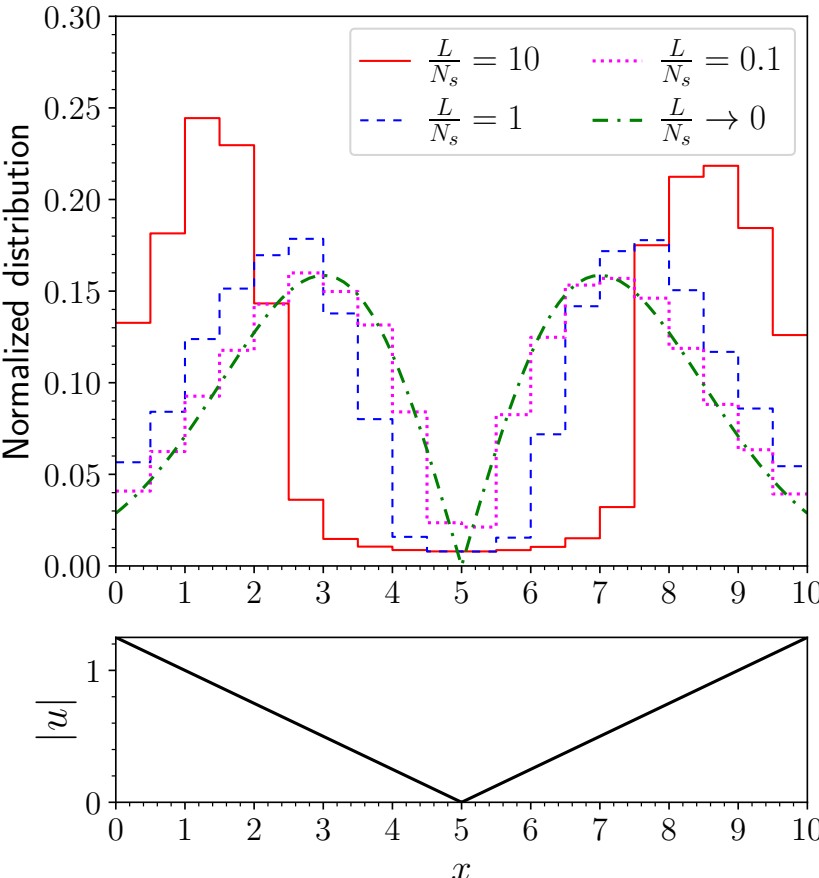

Figure 4: OASIS-trained distributions for several different choices of $L/N_s$ during training, as indicated by the legend. The green dot-dashed line shows the theoretical optimal sampling distribution in the large $N_s$ limit from (28) (the green dot-dashed line in Figure 1). The bottom panel shows the absolute value of the per-event score $|u|$ as a function of $x$.

considered in sections 2.2.3 and 2.2.4—we saw that when there is not enough simulated data, $N_s \ll LF$, the sampling distribution is focused entirely on the regions with the highest $|u|$ (similar to a delta function), while in the opposite limit, $N_s \gg LF$, the sampling distribution is more lenient towards lower magnitudes $|u|$ of the per-event score, with the weights simply being proportional to $|u|^{-1}$.

Having demonstrated the effect of OASIS on the histogram error-bars (i.e., the uncertainties in the differential cross-section estimated from the simulated data), we next turn to the effect of OASIS on the measurement of $\theta$. For this purpose, we perform a number of pseudo-experiments, where both real and simulated data are generated and compared to each other and the true value $\theta_{\text{true}}$ of the parameter $\theta$ is estimated by minimizing the $\chi^2$ statistic. Results from one representative pseudo-experiment are shown in the left panel of Figure 5, while the right panel of Figure 5 and Table 2 summarize the relevant results from the whole ensemble of pseudo-experiments.

Specifically, the data generation and the $\theta$ measurement were done as follows:

1. We performed two separate sets of pseudo-experiments. In each set, the integrated luminosity $L$ and number of simulated events $N_s$ were taken to be equal—$10,000$ for the first set and $100,000$ for the second set, as shown in the first two rows of Table 2.

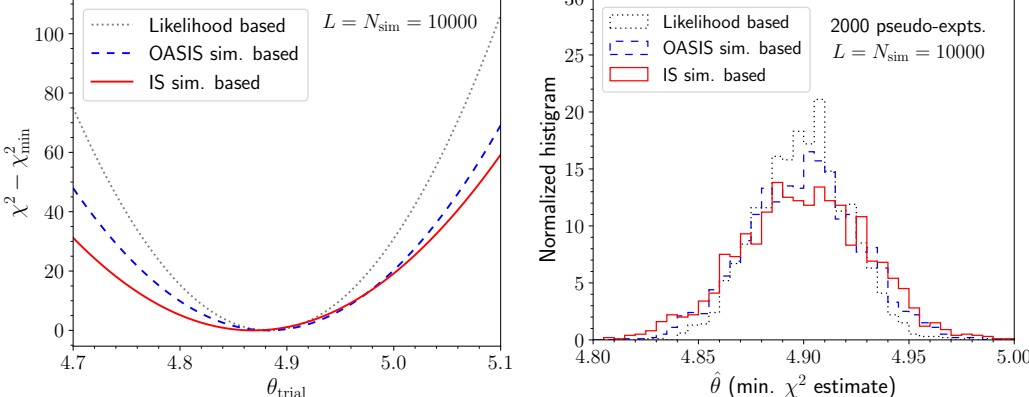

Figure 5: The left panel illustrates the minimum $\chi^2$ estimation of $\theta_{\text{true}}$ in one typical pseudo-experiment using a) the theoretical knowledge of the analytical form of (38) in order to construct the likelihood function (gray dotted line); b) simulation using OASIS-trained sampling distribution (blue dashed line); and c) simulation using IS sampling distribution (red solid line). The integrated luminosity $L$ for the real data, and the number of simulated events $N_s$ were both set to 10,000 (corresponding to roughly equal number of real and simulated events), and $\theta_{\text{true}} = 4.9$. The events were re-weighted for different $\theta_{\text{trial}}$ values as in [51, 52]. Higher convexity is indicative of lower error bars on the estimate $\hat{\theta}$. The right panel shows the distribution of $\hat{\theta}$ for the same three methods of $\theta$ estimation across 2000 pseudo-experiments, each with different instances of real and simulated datasets.

2. The underlying true value $\theta_{\text{true}}$ of the parameter $\theta$ was set to 4.9 (row 3 of Table 2) and the experimental data was generated as-per (38). The particular pseudo-experiment in the left panel of Figure 5 resulted in 9887 events (in our example, $F(\theta_{\text{true}}) \approx 0.9875$).

3. In each individual pseudo-experiment, two simulated data samples were produced. For the first one, we used the OASIS-trained distribution shown with the blue dashed line in the left panel of Figure 2, which was trained (only once) at $\theta_0 = 5$ with $L/N_s = 1$ (rows 4 and 5 of Table 2). The second simulated sample was produced under the ideal IS distribution which exactly matches $f/F$ (for consistency, we again used $\theta_0 = 5$). The simulated events were then re-weighted for different trial values $\theta_{\text{trial}}$ of the parameter $\theta$ [51, 52].

4. The real and simulated data were binned into 40 equally sized bins in the region $0 \leq x \leq 10$. A minimum $\chi^2$ estimation of $\theta_{\text{true}}$ was performed, accounting for statistical uncertainties in both the real and the simulated data [2], with the re-weighted simulations serving the role of theory expectation for different $\theta_{\text{trial}}$ values.

5. For comparison, we also perform the minimum $\chi^2$ estimation based on the likelihood function using the exact analytical expression (38). Note that this method does not suffer from *simulation* uncertainties and thus represents the ideal case reached in the infinite simulation statistics limit.

The left panel in Figure 5 illustrates the minimum $\chi^2$ estimation of $\theta_{\text{true}}$ in a typical pseudo-experiment from the set with $L = N_s = 10,000$, following each of the three methods described above, i.e., using a) the likelihood function formed with the exact analytical expression (38) (gray dotted line), b) the OASIS-trained sampling distribution (blue dashed line), and c) the IS sampling distribution (red solid line). The plot shows the value of the $\chi^2$ statistic (relative to

Table 2: Simulation parameters and summary statistics of the results from the simulated pseudo-experiments to measure $\theta_{\text{true}}$.

| | | | | | | |
|---|---|---|---|---|---|---|
| $L$ | 10, 000 | | | 100, 000 | | |
| $N_s$ | 10, 000 | | | 100, 000 | | |
| $\theta_{\text{true}}$ | 4.9 | | | 4.9 | | |
| Training $L/N_s$ | 1 | | | 1 | | |
| Simulation $\theta_0$ | 5 | | | 5 | | |
| Pseudo-expts. | 2000 | | | 500 | | |
| | ave. $\hat{\theta}$ | stdev $\hat{\theta}$ | $[\mathcal{I}_{\text{MC}}(\theta_{\text{true}})]^{-1/2}$ | ave. $\hat{\theta}$ | stdev $\hat{\theta}$ | $[\mathcal{I}_{\text{MC}}(\theta_{\text{true}})]^{-1/2}$ |
| Likelihood-based | 4.8997(5) | 2.15(3)E−2 | 2.108(1)E−2 | 4.9001(3) | 6.9(2)E−3 | 6.667(3)E−3 |
| OASIS-based | 4.9000(6) | 2.64(4)E−2 | 2.611(2)E−2 | 4.8998(4) | 8.5(3)E−3 | 8.258(5)E−3 |
| IS-based | 4.8999(7) | 3.03(5)E−2 | 2.957(19)E−2 | 4.9004(4) | 9.6(3)E−3 | 9.390(19)E−3 |

its minimum value $\chi^2_{\text{min}}$) as a function of the trial value $\theta_{\text{trial}}$ for $\theta$. As anticipated, in each case, the minimum $\chi^2$ is obtained in the vicinity of the true value $\theta_{\text{true}} = 4.9$, but the convexity of the function is different. This is important because the convexity near the minimum is indicative of the size of the error bars associated with the minimum $\chi^2$ estimate $\hat{\theta}$—higher convexity corresponds to lower error bars, and vice versa. We see that, as expected, the most precise measurement is offered by the ideal case when we use the analytical form of (38) and thus do not suffer from simulation uncertainties. At the same time, the comparison of the blue dashed and the red solid lines reveals that OASIS outperforms regular IS in terms of the precision on the $\hat{\theta}$ estimate, since the blue dashed curve is more convex that the red solid line.

In order to quantify the precision gains from using OASIS as opposed to regular IS, we analyze the results from the full ensemble of pseudo-experiments. The right panel shows the distribution of $\hat{\theta}$ values obtained in the 2000 pseudo-experiments (each with different instances of real and simulated datasets) performed with $L = N_s = 10000$, for each of the three methods: likelihood-based (grey dotted histogram), OASIS-simulation-based (blue dashed histogram), and IS-simulation-based (red solid histogram). The last three lines in Table 2 list the sample mean and standard deviation for $\hat{\theta}$, along with the square-root of the inverse of $\mathcal{I}_{\text{MC}}(\theta_{\text{true}})$ which is expected to be comparable to the total uncertainty. The histogram shapes in the right panel of Figure 5 confirm that OASIS outperforms IS and reduces the gap to the ideal sensitivity offered by the likelihood-based analysis. This can also be verified by inspecting the entries in Table 2 for the standard deviation of $\hat{\theta}$. Note that increasing the statistics to 100,000 events, as in the rightmost columns of Table 2, has the effect of reducing the measurement errors, but does not alter the performance rank of the three methods.

Note that the sensitivity of an experimental analysis will depend on the exact likelihood-free inference technique used, and in particular on how the theory expectations are estimated from the simulations. But regardless of the inference strategy, analyses will benefit from the preferential sampling of events in regions of higher sensitivity.

Table 2 also shows the values of $[\mathcal{I}_{\text{MC}}(\theta_{\text{true}})]^{-1/2}$ for the different simulation sampling distributions. For this purpose, $\mathcal{I}_{\text{MC}}/L$ is estimated using (17b) as the sample mean of $wu^2/(1+ Lw/N_s)$ (over 100,000 simulated events). $\mathcal{I}_{\text{MC}}$ for the likelihood-based case simply refers to $\mathcal{I}$, which can be estimated as the sample mean of $wu^2$ under either sampling distribution. Note the similarity between the estimated values of $[\mathcal{I}_{\text{MC}}(\theta_{\text{true}})]^{-1/2}$ and the corresponding standard deviations in $\hat{\theta}$. This establishes $\mathcal{I}_{\text{MC}}$ as a reliable measure of sensitivity offered by the respective simulated datasets[14].

---

[14]It is presently unclear whether $\mathcal{I}_{\text{MC}}$ provides a lower limit on the uncertainty of an appropriately defined,

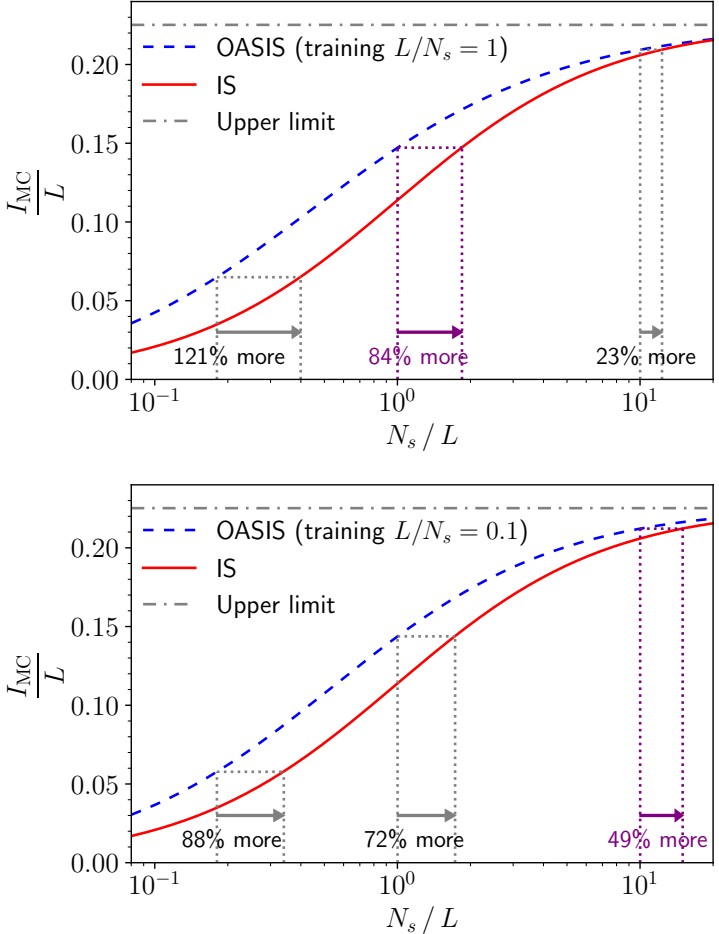

Figure 6: Dependence of $\mathcal{I}_{\mathrm{MC}}$ on $N_s/L$ for the OASIS-trained (blue dashed curve) and IS (red solid curve) sampling distributions. In the top (bottom) panel, the OASIS sampling distribution was trained at $L/N_s = 1$ ($L/N_s = 0.1$). The horizontal arrows indicate the percent reduction in the simulation requirements at three representative values of the ratio $N_s/L$. In both panels, the gray dot-dashed line indicates the theoretical upper bound obtained in the infinite simulated statistics limit.

Having established the relationship between the measurement uncertainty and $\mathcal{I}_{\mathrm{MC}}$, we will now use $\mathcal{I}_{\mathrm{MC}}/L$ at $\theta_0 = 5$ as a performance metric to quantify the resource conservation offered by OASIS. Figure 6 shows the value of $\mathcal{I}_{\mathrm{MC}}$ as a function of the available simulation statistics (as measured by the parameter $N_s/L$) for the OASIS-trained (blue dashed curve) and IS (red solid curve) sampling distributions. In each panel, the OASIS sampling distribution was only trained once: at $L/N_s = 1$ in the top panel and at $L/N_s = 0.1$ in the bottom panel. In spite of this, the same sampling distribution (already trained at a given fixed value of $L/N_s$) can still be reused to produce a different number of simulated events—it is the resulting dependence on $N_s$ which is depicted in Figure 6.

As can be seen from Figure 6, the OASIS-trained sampling distribution leads to higher values of $\mathcal{I}_{\mathrm{MC}}$ (and consequently, higher sensitivity of the analysis) for the same value of $N_s/L$. (The maximum achievable value of $\mathcal{I}_{\mathrm{MC}}/L$ is $\mathcal{I}/L$ (in the large $N_s$ limit), and it is also depicted as the gray dot-dashed horizontal line in the same figure.) Viewed differently, any target value of $\mathcal{I}_{\mathrm{MC}}$ is reached using fewer simulated events under the OASIS-trained sampling distribu-

---

generic likelihood-free estimator, like the Fisher information $\mathcal{I}$ does for a generic estimator of $\theta$.

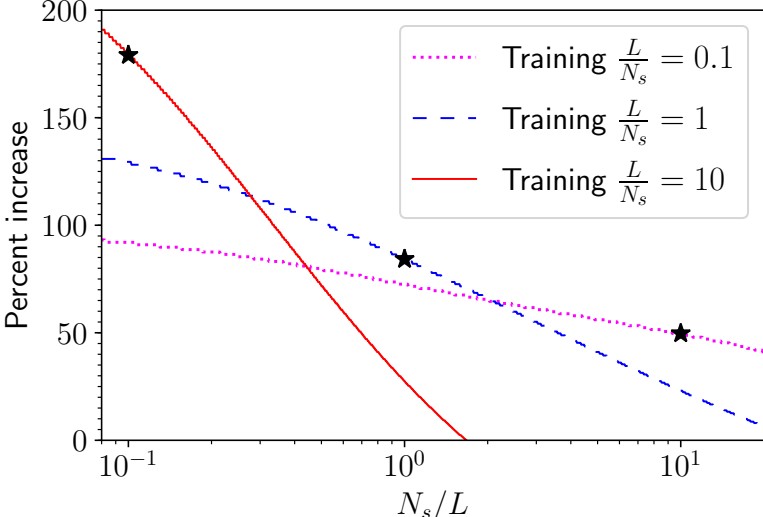

Figure 7: The percent increase (indicated by the horizontal arrows in Figure 6) in the required amount of simulated data to reach the same value of $\mathcal{I}_{\mathrm{MC}}/L$, if using regular IS instead of OASIS, as a function of $N_s/L$. Each ★ mark corresponds to the $N_s/L$ value a particular OASIS distribution was trained to be optimal at.

tion. This potential for resource conservation is depicted by the horizontal arrows in Figure 6, indicating the percent increase[15] in the number of simulated events that would have had to be generated under the regular IS scheme, for three different values of $N_s/L$. Figure 7 shows an alternate representation of this potential for resource conservation, by plotting the percent increase in the required amount of simulated data as a function of $N_s/L$ for the three different OASIS distributions trained at $L/N_s = 0.1$ (magenta dotted line), $L/N_s = 1$ (blue dashed line) and $L/N_s = 10$ (red solid line).

### 2.3.3 Deriving optimal "target weights" based on the score

The previous example demonstrates the construction of optimal sampling distributions when the phase space is one-dimensional. While the technique of adjusting the weights of cells (created by, say, the `Foam` algorithm) is applicable in higher dimensions as well, since the cells were not specifically created with OASIS in mind, the performance of the resulting sampling distribution may be limited.

In this section, we will convert the problem of OASIS-training the sampling distribution for a multi-dimensional phase space to a one-dimensional problem. The key observation is that for any sampling distribution $g$ (with weight function $w$), there exists a 'better' sampling distribution $g_{\mathrm{better}}$, with weights $w_{\mathrm{better}}$ that depend only on $|u|$, given by

$$g_{\mathrm{better}}(\boldsymbol{x}) = \frac{f(\boldsymbol{x})}{E_g\Big[\, w(\boldsymbol{x}) \,\Big|\, |u(\boldsymbol{x})|\, \Big]} \,, \tag{42}$$

$$w_{\mathrm{better}}(\boldsymbol{x}) = \frac{f(\boldsymbol{x})}{g_{\mathrm{better}}(\boldsymbol{x})} = E_g\Big[\, w(\boldsymbol{x}) \,\Big|\, |u(\boldsymbol{x})|\, \Big] \,, \tag{43}$$

---

[15]The reason why we plot the percent increase instead of the percent reduction is the following. For any given simulation budget $N_s$ read off from the $x$ axes of Figure 6 and Figure 7, one would naturally choose to use the more efficient sampling method, in which case the relevant question becomes, how much worse one would have been under the other, less efficient, method.

such that

$$\left\{\mathcal{I}_{\text{MC}} \text{ under } g_{\text{better}}\right\} \geq \left\{\mathcal{I}_{\text{MC}} \text{ under } g\right\} . \tag{44}$$

This can be intuitively understood from the special case considered in (22b)—it is better for the regions of phase space with the same value of $|u|$ to have the same weight, and $g_{\text{better}}$ simply redistributes the sampling distribution $g$ among regions of constant $|u|$ to make the weights the same. An explicit proof of (44) can be found in Appendix C. This means that there exists an optimal sampling distribution $g_{\text{optimal}}$ (for a given value of $\theta_0$), such that the weights of events under this distribution depend only on $|u(\boldsymbol{x})|$. If we can derive these optimal weights as a function of $|u|$, we can use them as target weights $w_{\text{target}}(|u(\boldsymbol{x})|)$ to be reached by the OASIS-trained sampling distribution. These target weights will correspond to a target sampling distribution

$$f_{\text{target}}(\boldsymbol{x}) = \frac{f(\boldsymbol{x})}{w_{\text{target}}(|u(\boldsymbol{x})|)} . \tag{45}$$

Note that $f_{\text{target}}$ can be computed using the oracle that provides $f(\boldsymbol{x})$ and $u(\boldsymbol{x})$, and a lookup table for the target weights. This allows us to employ existing 'regular' importance sampling techniques to train the sampling distribution $g$ to approximate the target sampling distribution $f_{\text{target}}$. Note that $f_{\text{target}}$ need not be normalized to 1, and the target weight function only needs to be learned up to a constant multiplicative factor for the approach to work.

To learn the functional form of $w_{\text{target}}(|u|)$, we will rewrite (17b) for sampling distributions of the form

$$g(\boldsymbol{x}) = \frac{f(\boldsymbol{x})}{w(|u(\boldsymbol{x})|)}, \tag{46}$$

as

$$\frac{\mathcal{I}_{\text{MC}}}{L} = \int d\boldsymbol{x} \, f(\boldsymbol{x}) \, \frac{u^2(\boldsymbol{x})}{1 + \dfrac{L}{N_s} w(|u(\boldsymbol{x})|)} \tag{47a}$$

$$= \int d|u| \, f_{|u|}(|u|) \, \frac{u^2}{1 + \dfrac{L}{N_s} w(|u|)} \tag{47b}$$

$$= \int d|u| \, f_{|u|}(|u|) \, \frac{u^2}{1 + \dfrac{L}{N_s} \dfrac{f_{|u|}(|u|)}{g_{|u|}(|u|)}} , \tag{47c}$$

where $f_{|u|}$ and $g_{|u|}$ are the distributions of $|u(\boldsymbol{x})|$ under the distributions $f$ and $g$ for $\boldsymbol{x}$, and in arriving at (47c) we have used

$$w(|u|') = \frac{f(\boldsymbol{x})}{g(\boldsymbol{x})}\bigg|_{|u(\boldsymbol{x})|=|u|'} = \frac{\displaystyle\int d\boldsymbol{x} \, f(\boldsymbol{x}) \, \delta_D\big(|u(\boldsymbol{x})|-|u|'\big)}{\displaystyle\int d\boldsymbol{x} \, g(\boldsymbol{x}) \, \delta_D\big(|u(\boldsymbol{x})|-|u|'\big)} = \frac{f_{|u|}(|u|)}{g_{|u|}(|u|)} , \tag{48}$$

where $\delta_D$ is the Dirac delta function. Comparing (47c) with (17b), we can see that the problem of deriving the optimal weights as a function of $|u|$ is identical to the problem of deriving optimal weights as a function of $\boldsymbol{x}$ which was tackled in the previous section. The only difference is that we do not have an oracle to return the value of $f_{|u|}(|u(\boldsymbol{x})|)$ for every generated event $\boldsymbol{x}$. But because this is just a one-dimensional problem, we can easily estimate the distribution $f_{|u|}$ using simulations, possibly from a different sampling distribution.

### 2.3.4 Direct construction using machine learning

Recently, there has been a significant interest in employing generative neural networks to perform importance sampling [41–46]. The idea is as follows: The neural network takes in as input a random vector $r$, sampled from a given known distribution $P_r(r)$ (e.g., multi-variate standard normal), and maps it to the output vector $x(r)$. The weights of the neural network, say $\varphi$, parameterize the map from $r$ to $x$, and this map governs the sampling distribution $g$ of $x$. Now, if the sampling distribution $g(x)$ a) covers the support of the true distribution $f$ (phase space where $f$ is non-zero), and b) can be computed for every sampled event $x$, then the neural network can be used to perform importance sampling. If the neural network architecture is chosen to be manifestly one-to-one, then $g(x(r))$ can be computed as

$$g(x(r)) = \frac{P_r(r)}{J} \, , \tag{49}$$

where $J \equiv |\nabla_r x|$ is the determinant of the Jacobian matrix of the map induced by the neural network. As a reminder, the weights of the generated events will be computed as $w = f/g$.

Such neural networks can be trained using gradient descent to maximize $\mathcal{I}_{\text{MC}}$ (or minimize $-\mathcal{I}_{\text{MC}}$). From (17b) we have

$$-\mathcal{I}_{\text{MC}} = \int dx \, f(x) \, \frac{-u^2(x)}{1 + \dfrac{L}{N_s} \dfrac{f(x)}{g(x)}} \, , \tag{50}$$

$$\Rightarrow -\nabla_\varphi \mathcal{I}_{\text{MC}} = \int dx \, f(x) \, \frac{-u^2(x)}{\left[1 + \dfrac{L}{N_s} \dfrac{f(x)}{g(x)}\right]^2} \, \frac{f(x)}{g(x)} \left[\nabla_\varphi \ln g(x)\right] \tag{51a}$$

$$= \int dx \, g(x) \, \frac{-w^2(x) u^2(x)}{\left[1 + \dfrac{L}{N_s} w(x)\right]^2} \left[\nabla_\varphi \ln g(x)\right] \tag{51b}$$

$$= E_g \left[-\left(\frac{wu}{1 + Lw/N_s}\right)^2 \left[\nabla_\varphi \ln g\right]\right] . \tag{51c}$$

This expression for the gradient of $-\mathcal{I}_{\text{MC}}$ as an expectation over events sampled as-per $g(x)$ facilitates the use of stochastic or (mini)-batch gradient descent to train the neural network, similar to the importance sampling loss functions in Refs. [43, 44, 46]. The viability of training generative networks using our loss function (or other related surrogate loss functions) is not explored in this work; here we merely intend to communicate the possibility to the community.

## 3 OASIS for analysis variables

### 3.1 Groundwork

The advantage of the parton-level event specification in MC simulations is that the probability distributions of parton-level variables under a given theory model are exactly computable by an oracle. This is why HEP simulations begin with the parton-level Monte Carlo, followed by the remaining stages of the simulation chain. The methods developed in Section 2 relied on the oracle to compute $f(x)$ and $u(x)$, and on the fact that the weight of an event is uniquely determined by its $x$ value. In this section we will develop the framework for applying OASIS at the analysis level, accounting for the following experimental realities:

- Only *reconstructed* versions of visible parton-level final state particles are available to the analyses.

- The kinematic information about *invisible* final state particles (such as neutrinos or dark matter candidates) is inaccessible.

- At hadron colliders, the particle ids and momentum fractions of the incoming partons in a given event are a priori unknown.

- Typically all of the available event information is reduced to a handful of sensitive event variables on which the analyses is performed.

- The analysis only uses events which pass the trigger requirements, the event selection criteria and the analysis cuts. In addition, the remaining events could be partitioned into several different categories to be analyzed separately.

To take these into account, let $v$ represent the (possibly multi-dimensional) event variable in the final analysis corresponding to an event. A parton-level event $x$ is mapped to $v$ in a probabilistic many-to-many manner, via the rest of the simulation pipeline, the reconstruction pipeline, and the event variable calculation. We will use $v$ to capture all the information used by the analysis, including any event selection or categorization information—if a particular event does not meet the selection cuts, $v$ can carry a special `Null` or `rejected` tag, and if the events are split into different categories (based on purity, for example), the category information could be included as a dimension in $v$. Let the transfer function $\text{TF}(v \,|\, x)$ represent the normalized distribution of $v$ conditional on $x$. Let $\mathcal{F}(v \,;\, \theta)$ and $\mathcal{G}(v)$ be the distributions of $v$ corresponding to the parton-level distributions $f(x \,;\, \theta)$ and $g(x)$ after marginalizing over $x$ using the transfer function TF:

$$\mathcal{G}(v) = \int dx \ g(x) \, \text{TF}(v \,|\, x) \,, \tag{52}$$

$$\mathcal{F}(v \,;\, \theta) = \int dx \ f(x \,;\, \theta) \, \text{TF}(v \,|\, x) \tag{53a}$$

$$= \int dx \ g(x) \, \text{TF}(v \,|\, x) \, w(x \,;\, \theta) \tag{53b}$$

$$= \int dx \ \mathcal{G}(v) \, \text{ITF}_g(x \,|\, v) \, w(x \,;\, \theta) \tag{53c}$$

$$= \mathcal{G}(v) \, E_g[w \,|\, v \,;\, \theta] \,, \tag{53d}$$

where the inverse transfer function $\text{ITF}_g(x \,|\, v)$ represents the normalized distribution of $x$ conditional on $v$, when the prior distribution on $x$ is $g$. Just like their parton-level counterparts, $\mathcal{F}(v \,;\, \theta)$ and $\mathcal{G}(v)$ are normalized to $F(\theta)$ and 1, respectively.

For an analysis performed on $v$, the relevant Fisher information is given by

$$\mathcal{I}(\theta_0) = \int dv \ \frac{1}{L \, \mathcal{F}(v \,;\, \theta_0)} \left[ L \, \frac{\partial \mathcal{F}(v \,;\, \theta)}{\partial \theta} \right]^2 \Bigg|_{\theta = \theta_0} \,, \tag{54}$$

where the integral is performed only over the non-`Null` or non-`rejected` values of $v$. Proceeding as before, the expectation and variance for the number of real events in a bin of size $\Delta v$ (small) at a given value of $v$, for a given value of $\theta = \theta_0$ is given by $L \, \mathcal{F}(v \,;\, \theta_0) \Delta v$. The simulated estimate for the expected event-count, as per (53d), is given by the sum of weights of the simulated events in the relevant bin, say $S_w$ (a random variable), scaled by a factor of

$L/N_s$. To estimate the variance of $S_w$, note that $S_w$ can be expressed as the sum of $N_s$ independent random variables $A_i \equiv I_i \times B_i$, where $B_i$-s are independent random variables following the same distribution as the weights of simulated events in the bin and $I_i$-s are indicator random variables which take value 1 with probability $\mathcal{G}(\boldsymbol{v})\Delta\boldsymbol{v}$ and 0 otherwise. Now,

$$S_w = \sum_{i=1}^{N_s} A_i = \sum_{i=1}^{N_s} I_i \times B_i \tag{55}$$

$$E_g[S_w] = N_s \times E_g[I_i] \times E_g[B_i] \tag{56a}$$

$$= N_s \times \mathcal{G}(\boldsymbol{v})\Delta\boldsymbol{v} \times E_g[w\,|\,\boldsymbol{v}\,;\,\theta_0] \tag{56b}$$

$$= N_s \mathcal{F}(\boldsymbol{v}\,;\,\theta_0)\Delta\boldsymbol{v}\,, \tag{56c}$$

$$\Rightarrow E\left[S_w \times \frac{L}{N_s}\right] = L\,\mathcal{F}(\boldsymbol{v}\,;\,\theta_0)\Delta\boldsymbol{v} \tag{57}$$

$$\mathrm{var}_g[S_w] = N_s\,\mathrm{var}_g[A_i] \tag{58a}$$

$$= N_s\left[E_g\left[I_i^2\right]\,E_g\left[B_i^2\right] - E_g[I_i]^2\,E_g[B_i]^2\right] \tag{58b}$$

$$= N_s\left[\mathcal{G}(\boldsymbol{v})\Delta\boldsymbol{v}\,E_g[w^2\,|\,\boldsymbol{v}\,;\,\theta_0] + \mathcal{O}\big((\Delta\boldsymbol{v})^2\big)\right] \tag{58c}$$

$$\Rightarrow \mathrm{var}_g\left[S_w \times \frac{L}{N_s}\right] = \frac{L^2}{N_s}\mathcal{G}(\boldsymbol{v})E_g[w^2\,|\,\boldsymbol{v}\,;\,\theta_0]\Delta\boldsymbol{v} + \mathcal{O}\big((\Delta\boldsymbol{v})^2\big) \tag{59a}$$

$$= \frac{L}{N_s}\frac{E_g[w^2\,|\,\boldsymbol{v}\,;\,\theta_0]}{E_g[w\,|\,\boldsymbol{v}\,;\,\theta_0]} \times L\,\mathcal{F}(\boldsymbol{v}\,;\,\theta_0)\Delta\boldsymbol{v} + \mathcal{O}\big((\Delta\boldsymbol{v})^2\big)\,. \tag{59b}$$

In (56a) and (58b), we have used the fact that $I$ and $B$ are independent by construction. In (56b) and (58c) we have used $E_g[I] = E_g[I^2] = \mathcal{G}(\boldsymbol{v})\Delta\boldsymbol{v}$ (the binomial 'success' probability), and in (59b), we have used (53d). Equation (58c) is related to the well-known formula for error-bars in unnormalized weighted histograms, given by the square root of the sum of squares of the weights in a given bin [63].

Introducing the uncertainty in $S_w L/N_s$ from (59b) into the expression for Fisher information (after dropping the $\mathcal{O}\big((\Delta\boldsymbol{v})^2\big)$ term), we get

$$\mathcal{I}_{\mathrm{MC}}(\theta_0) = \int d\boldsymbol{v}\,\frac{\left[L\dfrac{\partial\mathcal{F}(\boldsymbol{v}\,;\,\theta)}{\partial\theta}\right]^2\Bigg|_{\theta=\theta_0}}{L\,\mathcal{F}(\boldsymbol{v}\,;\,\theta_0)\left[1+\dfrac{L}{N_s}\dfrac{E_g[w^2\,|\,\boldsymbol{v}\,;\,\theta_0]}{E_g[w\,|\,\boldsymbol{v}\,;\,\theta_0]}\right]}\,, \tag{60}$$

which is the analogue of (16). Further, in analogy to (17), this result can be rewritten as

$$\frac{\mathcal{I}_{\mathrm{MC}}(\theta_0)}{L} = \int d\boldsymbol{v}\,\mathcal{F}(\boldsymbol{v}\,;\,\theta_0)\,\frac{\left[\partial_\theta\ln[\mathcal{F}(\boldsymbol{v}\,;\,\theta)]\right]^2\Big|_{\theta=\theta_0}}{1+\dfrac{L}{N_s}\dfrac{E_g[w^2\,|\,\boldsymbol{v}\,;\,\theta_0]}{E_g[w\,|\,\boldsymbol{v}\,;\,\theta_0]}} \tag{61a}$$

$$= \int d\boldsymbol{v}\,\mathcal{F}(\boldsymbol{v}\,;\,\theta_0)\,\frac{\mathcal{U}^2(\boldsymbol{v}\,;\,\theta_0)}{1+\dfrac{L}{N_s}\dfrac{E_g[w^2\,|\,\boldsymbol{v}\,;\,\theta_0]}{E_g[w\,|\,\boldsymbol{v}\,;\,\theta_0]}}\,, \tag{61b}$$

where $\mathcal{U}(\boldsymbol{v}\,;\,\theta_0)$ is the per-event score at the analysis level given by

$$\mathcal{U}(\boldsymbol{v}\,;\,\theta) \equiv [\partial_\theta\ln[\mathcal{F}(\boldsymbol{v}\,;\,\theta)]] = \left[\frac{1}{\mathcal{F}(\boldsymbol{v}\,;\,\theta)}\frac{\partial\mathcal{F}(\boldsymbol{v}\,;\,\theta)}{\partial\theta}\right]\,. \tag{62}$$

As before, for notational convenience, we will suppress the $\theta_0$ dependence in the different distributions and quantities, unless deemed useful. Typically the MC dataset for a given analysis is composed of several background and signal subsamples. Furthermore, there could be systematic uncertainties in the MC as well, both from the theory side (e.g., neglecting higher order corrections, working in the narrow-width approximation, etc.) and from the experiment side. All of these can be incorporated into (61b) in a straightforward manner as

$$\frac{\mathcal{I}_{\mathrm{MC}}}{L} = \int d\boldsymbol{v} \, \mathcal{F}(\boldsymbol{v}) \, \frac{\mathcal{U}^2(\boldsymbol{v})}{1 + \dfrac{\sigma_{\mathrm{syst}}^2(\boldsymbol{v})}{\sigma_{\mathrm{stat}}^2(\boldsymbol{v})} + \sum_k \dfrac{\mathcal{F}^{(k)}(\boldsymbol{v})}{\mathcal{F}(\boldsymbol{v})} \dfrac{L}{N_s^{(k)}} \dfrac{E_{g^{(k)}}[w_k^2 \,|\, \boldsymbol{v}]}{E_{g^{(k)}}[w_k \,|\, \boldsymbol{v}]}} \,, \tag{63}$$

where $\sigma_{\mathrm{syst}}$ is the systematic uncertainty (in the relevant bin at $\boldsymbol{v}$) in the MC data unrelated to its finiteness, $\sigma_{\mathrm{stat}}$ is the statistical uncertainty (in the relevant bin at $\boldsymbol{v}$) in the real dataset[16], and $k$ is a subsample index. $N_s^{(k)}$ is the number of simulated events in the $k$-th subsample with $N_s = \sum_k N_s^{(k)}$ being the total number of simulated events. Each subsample has its own true distribution $f^{(k)}$ (normalized to the cross-section of the $k$-th process) and sampling distribution $g^{(k)}$ (normalized to 1). The weight $w_k(\boldsymbol{x})$ of a parton-level event $\boldsymbol{x}$ in subsample $k$ is given by $f^{(k)}(\boldsymbol{x})/g^{(k)}(\boldsymbol{x})$. $\mathcal{F}^{(k)}(\boldsymbol{v})$ and $\mathcal{F}(\boldsymbol{v})$ are defined by

$$\mathcal{F}^{(k)}(\boldsymbol{v}) \equiv \int d\boldsymbol{x} \, f^{(k)}(\boldsymbol{x}) \, \mathrm{TF}(\boldsymbol{v} \,|\, \boldsymbol{x}) = \mathcal{G}^{(k)}(\boldsymbol{v}) \, E_{g^{(k)}}[w_k \,|\, \boldsymbol{v}] \,, \tag{64}$$

$$\mathcal{F}(\boldsymbol{v}) = \sum_k \mathcal{F}^{(k)}(\boldsymbol{v}) \,, \tag{65}$$

where $\mathcal{G}^{(k)}(\boldsymbol{v})$ is given by

$$\mathcal{G}^{(k)}(\boldsymbol{v}) \equiv \int d\boldsymbol{x} \, g^{(k)}(\boldsymbol{x}) \, \mathrm{TF}(\boldsymbol{v} \,|\, \boldsymbol{x}) \,. \tag{66}$$

Note that reducible backgrounds as well as backgrounds with different sets of invisible final state particles will live in different spaces at the parton-level.

At first sight, the task of deriving good sampling distributions $g^{(k)}$ (in their respective domains) using (63) seems like a daunting task considering that the terms in the expression cannot be exactly computed and live in the realm of individual analyses, while $g^{(k)}$ live in the parton-level MC realm. However, the following observations facilitate the task at hand:

1. **The relevant quantities in** (63) **can be estimated from a preliminary smaller MC sample.** $\mathcal{F}$, $\mathcal{F}^{(k)}$, $\mathcal{U}$, and $\sigma_{\mathrm{syst}}/\sigma_{\mathrm{stat}}$ can be estimated using 'preliminary' (possibly pre-existing) simulated datasets much smaller than the final MC dataset. For example, $\sigma_{\mathrm{stat}}$ in a given bin can be extrapolated from a smaller MC dataset, with $\sigma_{\mathrm{syst}}$ being independent of the size of the dataset. Similarly, $\mathcal{U}$ can be estimated either from the estimate of $\mathcal{F}$ for neighboring values of $\theta$, or more directly as the appropriately weighted average of $u(\boldsymbol{x})$ (or equivalently $\partial_\theta \ln w$), conditional on $\boldsymbol{v}$ [64]. For example, for a single component sample, from (62) and (53d) we have

$$\mathcal{U}(\boldsymbol{v}) = \frac{1}{\mathcal{F}(\boldsymbol{v})} \frac{\partial \mathcal{F}(\boldsymbol{v})}{\partial \theta} = \frac{E_g[\partial_\theta w \,|\, \boldsymbol{v}]}{E_g[w \,|\, \boldsymbol{v}]} \tag{67a}$$

$$= \frac{E_g[w \, \partial_\theta \ln w \,|\, \boldsymbol{v}]}{E_g[w \,|\, \boldsymbol{v}]} = \frac{E_g[w u \,|\, \boldsymbol{v}]}{E_g[w \,|\, \boldsymbol{v}]} \tag{67b}$$

$$= \frac{E_f[(w/w) u \,|\, \boldsymbol{v}]}{E_f[w/w \,|\, \boldsymbol{v}]} = E_f[u \,|\, \boldsymbol{v}] \,. \tag{67c}$$

---

[16]Note that while $\sigma_{\mathrm{stat}}$ and $\sigma_{\mathrm{syst}}$ individually depend on the bin-width, their ratio does not (up to local smoothing effects). The ratio is also independent of whether the $\sigma$-s refer to absolute or relative uncertainties.

Similarly, for the case with multiple components, from (62), (65), and (67) we have

$$\mathcal{U}(\boldsymbol{v}) = \partial_\theta \ln \mathcal{F}(\boldsymbol{v}) = \sum_k \frac{\mathcal{F}^{(k)}(\boldsymbol{v})}{\mathcal{F}(\boldsymbol{v})} \, \partial_\theta \ln \mathcal{F}^{(k)}(\boldsymbol{v}) \tag{68a}$$

$$= \sum_k \frac{\mathcal{F}^{(k)}(\boldsymbol{v})}{\mathcal{F}(\boldsymbol{v})} \frac{E_{g^{(k)}}[w_k u_k | \boldsymbol{v}]}{E_{g^{(k)}}[w_k | \boldsymbol{v}]} \tag{68b}$$

$$= \sum_k \frac{\mathcal{F}^{(k)}(\boldsymbol{v})}{\mathcal{F}(\boldsymbol{v})} \, E_{f^{(k)}}[u_k | \boldsymbol{v}] \,, \tag{68c}$$

where $u_k$ is simply $\partial_\theta w_k$. Note that although the estimates of $\mathcal{F}, \mathcal{F}^{(k)}, \mathcal{U}$, and $\sigma_{\text{syst}}/\sigma_{\text{stat}}$ from a preliminary dataset will not be accurate up to the sensitivity offered by the full MC dataset [50], they will be sufficient for projecting, with sufficient accuracy, the sensitivity of the experiment under different sampling distributions.

2. **Target weights in the $\boldsymbol{v}$ space can be translated to weights in the $\boldsymbol{x}$ space.** Although the map from $\boldsymbol{x}$ to $\boldsymbol{v}$ is technically many-to-many, it is usually approximately many-to-one, with the event variable $\boldsymbol{v}$ for a given value of the parton-level event $\boldsymbol{x}$ typically falling within a small window of possibilities. This means that if the individual analyses can identify 'target' sampling weights for different regions in $\boldsymbol{v}$, it can then be translated to weights in the $\boldsymbol{x}$ (parton-level) phase space.

Here we are proposing to restrict our attention to sampling distributions which (roughly) assign the same weights to all $\boldsymbol{x}$ values which (roughly) map to the same value of $\boldsymbol{v}$. This can be justified as follows. Since

$$E_{g^{(k)}}[w_k^2 | \boldsymbol{v}] \geq \left[ E_{g^{(k)}}[w_k | \boldsymbol{v}] \right]^2 \,, \tag{69}$$

it follows from (63) that

$$\left\{ \frac{\mathcal{I}_{\text{MC}}}{L} \text{ under } \{g^{(k)}\} \right\} \leq \int d\boldsymbol{v} \; \frac{\mathcal{F}(\boldsymbol{v}) \, \mathcal{U}^2(\boldsymbol{v})}{1 + \dfrac{\sigma_{\text{syst}}^2(\boldsymbol{v})}{\sigma_{\text{stat}}^2(\boldsymbol{v})} + \sum_k \dfrac{\mathcal{F}^{(k)}(\boldsymbol{v})}{\mathcal{F}(\boldsymbol{v})} \dfrac{L}{N_s^{(k)}} E_{g^{(k)}}[w_k | \boldsymbol{v}]} \,, \tag{70}$$

with equality iff $E_{g^{(k)}}[w_k^2 | \boldsymbol{v}] = \left[ E_{g^{(k)}}[w_k | \boldsymbol{v}] \right]^2$ (i.e., if the variance of $w$ within subsample $k$, conditional on $\boldsymbol{v}$ is 0) for all $k$ and almost all $\boldsymbol{v}$. If the map from $\boldsymbol{x}$ to $\boldsymbol{v}$ is strictly deterministic (many-to-one), then from a given set of sampling distributions $\{g^{(k)}\}$, we can construct a better set of sampling distributions $\{g_{\text{better}}^{(k)}(\boldsymbol{x})\}$, with weights $w_{k,\text{better}}$ which only depend on $\boldsymbol{v}(\boldsymbol{x})$, given by

$$g_{\text{better}}^{(k)}(\boldsymbol{x}) = \frac{f^{(k)}(\boldsymbol{x})}{E_{g^{(k)}}[w_k | \boldsymbol{v}(\boldsymbol{x})]} \,, \tag{71}$$

$$w_{k,\text{better}}(\boldsymbol{x}) = \frac{f^k(\boldsymbol{x})}{g_{\text{better}}^{(k)}(\boldsymbol{x})} = E_{g^{(k)}}[w_k | \boldsymbol{v}(\boldsymbol{x})] \,, \tag{72}$$

such that the value of $\mathcal{I}_{\text{MC}}$ under $\{g_{\text{better}}^{(k)}\}$ is greater than or equal to that under $\{g^{(k)}\}$. This is because the right-hand-side of (70) equals the value of $\mathcal{I}_{\text{MC}}/L$ under $\{g_{\text{better}}^{(k)}\}$ (based on the expression in (63)), since

$$E_{g_{\text{better}}^{(k)}} \left[ w_{k,\text{better}}^2 | \boldsymbol{v} \right] = \left[ E_{g_{\text{better}}^{(k)}} [w_{k,\text{better}} | \boldsymbol{v}] \right]^2 = \left[ E_{g^{(k)}}[w_k | \boldsymbol{v}] \right]^2 \,. \tag{73}$$

These two observations (the statements in boldface in items 1 and 2 above) lead to a two-stage approach to performing OASIS for a realistic analysis, which we will describe next.

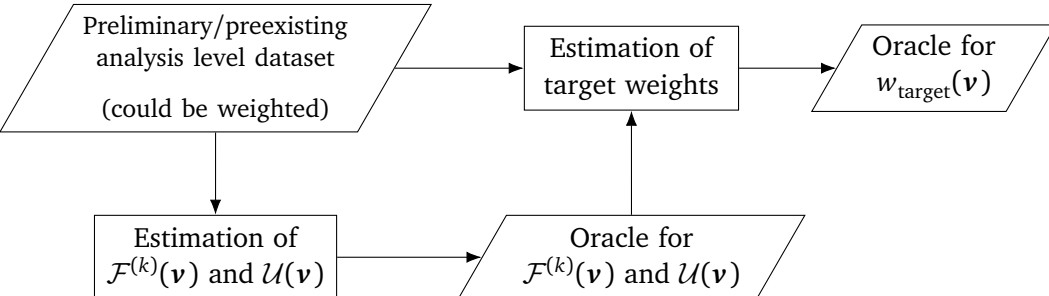

Figure 8: A flowchart of the main steps of the first stage of OASIS discussed in Section 3.2.1. Rectangles refer to processes or actions, while the remaining parallelograms refer to the corresponding inputs or deliverables. The estimation of $\mathcal{F}^{(k)}(\boldsymbol{v})$ and $\mathcal{U}(\boldsymbol{v})$ can be done using existing regression techniques. The techniques introduced in Section 2 can be used to perform the estimation of $w_{\text{target}}(\boldsymbol{v})$ by maximizing (75).

## 3.2 Constructing optimal sampling distributions

### 3.2.1 Stage 1: Taking stock at the analysis level

Following up on the second observation above, let us restrict our attention to sampling distributions of the form

$$g^{(k)}(\boldsymbol{x}) = \frac{f^{(k)}(\boldsymbol{x})}{w_k(\boldsymbol{v}(\boldsymbol{x}))} \ . \tag{74}$$

For this case, (63) reduces to

$$\frac{\mathcal{I}_{\text{MC}}}{L} = \int d\boldsymbol{v} \ \mathcal{F}(\boldsymbol{v}) \ \frac{\mathcal{U}^2(\boldsymbol{v})}{1 + \dfrac{\sigma_{\text{syst}}^2(\boldsymbol{v})}{\sigma_{\text{stat}}^2(\boldsymbol{v})} + \sum_k \dfrac{\mathcal{F}^{(k)}(\boldsymbol{v})}{\mathcal{F}(\boldsymbol{v})} \dfrac{L}{N_s^{(k)}} w_k(\boldsymbol{v})} \ . \tag{75}$$

Since all the terms in the right hand side of (75) other than $w_k$ are either heuristic parameters or calculable using a small preliminary MC dataset, we can estimate good target weights $w_{k,\text{target}}(\boldsymbol{v})$ for different values of $\boldsymbol{v}$ in the different subsamples $k$ using the same technique as was used to maximize the expression in (17b) in this paper.

The utility of different regions is captured by the $\mathcal{U}^2$ term in the numerator. This can be appropriately substituted for purposes other than parameter estimation. For example, in a signal search analysis it can be replaced by $(s/b)^2$, since $\mathcal{U}$ is proportional to the signal-to-background ratio when $\theta$ is the signal cross-section and the sensitivity is to be maximized around $\theta = 0$. If simulated events are needed in a nonsensitive control region for MC validation, the term can be fixed to an appropriate value by hand. Furthermore, if the optimization performed in estimating the target weights $w_{k,\text{target}}$ is deemed too aggressive, one can adjust the weights appropriately by hand. In short, at this stage individual analyses can take stock and identify a preferred MC sampling distribution in the phase space of the event variable $\boldsymbol{v}$ for each of the relevant subsamples. The flowchart in Figure 8 summarizes the main steps of this stage of OASIS.

### 3.2.2 Stage 2: Translating the target weights to parton-level

In this stage we will choose the parton-level sampling distribution $g$ in the phase space of $\boldsymbol{x}$ to approach the desired target weights in $\boldsymbol{v}$. Since we are only considering the generation of one process at a time, we will drop the subsample index $k$ for notational convenience.

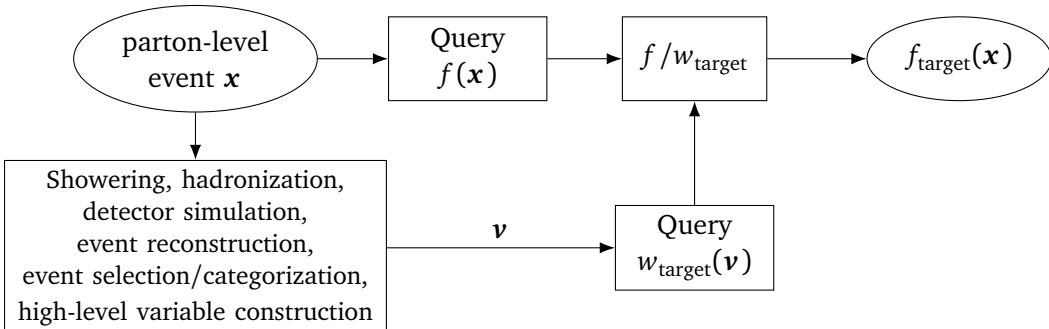

Figure 9: A flowchart of the oracle for $f_{\text{target}}$ to be used in the second stage of OASIS discussed in Section 3.2.2. The query for $f(x)$ can be performed with the existing standard HEP machinery, while the query for $w_{\text{target}}(v)$ can be done with the oracle trained in Section 3.2.1.

For this purpose, we can employ an existing importance sampling approach like the VEGAS algorithm [35,36], the Foam algorithm [37–40], or a neural network based approach [41–46]. Each of these methods needs to be provided with an oracle that can be queried for the value of the underlying (possibly unnormalized) distribution we are trying to sample from. Usually this oracle simply returns $f(x)$, which is a combination of parton distribution functions and the relevant matrix elements.

In our case, in addition to $f$, the oracle will be based on a simulation and reconstruction pipeline to transform $x$ into $v$, and the target weights $w_{\text{target}}$ for different $v$ values produced in stage 1. For each queried event $x$, the oracle will run the simulation forward to find an associated target weight, and return $f_{\text{target}} \equiv f(x)/w_{\text{target}}$. The flowchart in Figure 9 depicts the internals of the oracle for $f_{\text{target}}(x)$. Note that much of the existing standard HEP simulation tools can be re-purposed for our analysis, in particular the query for $f(x)$, the production of $v$ from $x$ and the existing importance sampling algorithms to mimic $f_{\text{target}}$.

In concluding this subsection, we provide the following usage notes.

- Since the map from $x$ to $v$ is not deterministic, the oracle may return different values of $f_{\text{target}}$ for the same query $x$. However, the existing importance sampling algorithms are robust under this non-determinism and will simply settle on an intermediate $f_{\text{target}}$.[17]

  This way, even if a region in the $x$ space only contributes rarely to an important or sensitive region in the $v$ space, this contribution will be taken into account by our approach, and the $x$-region will receive a suitably high sampling rate.

- If the event $x$ (after forward smearing to $v$) does not pass the selection thresholds of the analysis, its target weight can be set to $\infty$ since it holds no utility for the analysis. However, this may be too aggressive, and using a suitably high maximum target weight maybe more appropriate.

  For example, there may be a signal region with contributions from reducible backgrounds due to very rare (in the relative sense) fakes or fluctuations in a process with a large rate. In such cases, a smaller preliminary dataset might not show any contributions from this background component, but manually setting an upper limit on the target weight will ensure that these regions receive adequate coverage by the sampling distribution.

---

[17]An equivalent compromise is incorporated into the OASIS results in Section 2.3.2 by the use of wide bins where $g$ is constrained to be constant, thus limiting the flexibility and forcing a compromise.

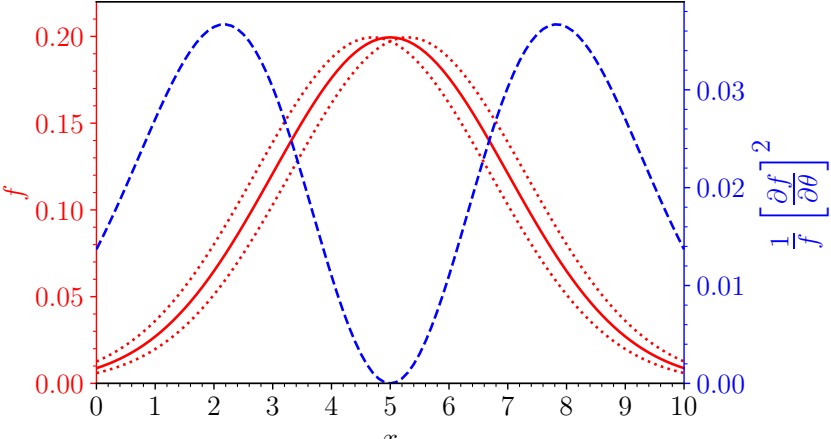

Figure 10: The three distributions of $f(x)$ from Figure 1 (red solid and dotted lines, left $y$-axis) and the distribution of the local shape sensitivity (77) (blue dashed line, right $y$-axis) for our toy example.

- Computationally inexpensive fast simulators may be sufficient for the purpose of mapping $x$ to event variables $v$ in Figure 9, since the goal is only to attain a good sampling distribution $g$.

- The $f_{\text{target}}$ oracle is to be used only in the training phase of the importance sampling algorithm. When actually generating events for the analysis, the weights for the sampled events should be based on the true distribution $f(x)$ as $w(x) = f(x)/g(x)$.

- Note that the individual $L/N_s^{(k)}$ values in (75) can also be optimized [65]. For example, if events from different sub-channels are equally costly to produce and process, the relevant constraint on $N_s^{(k)}$ is simply

$$N_s = \sum_k N_s^{(k)} \,. \tag{76}$$

On the other hand, if events from different sub-channels have different computational costs, the above constraint can be modified using suitable weights for the individual terms.

- If the same sample of simulated events will be used in multiple analyses, a common ground sampling distribution should be found by taking into account the requirements of the individual analyses.

### 3.3 Comparison to a real-life example from a CMS analysis

To appreciate the potential for resource conservation offered by OASIS, let us consider the measurement of the top quark mass by the Compact Muon Solenoid (CMS) experiment in the dileptonic $t\bar{t}$ decay channel presented in [66]. Fig. 1b (available here), Fig. 3b (available here), and Fig. 4b (available here) of that publication show distributions of different event variables ($M_{bl}$, $M_{T2}^{bb}$, and $M_{bl\nu}$, respectively) for three nearby values of the top quark mass $M_t^{MC}$ used in the MC: 166.5 GeV, 172.5 GeV and 178.5 GeV. This is analogous to the comparison between the three red curves in Figure 1 (one solid and two dotted, reproduced again in Figure 10), where the role of $\theta$ is played by $M_t^{MC}$. By comparing the shapes of the curves at different values of $M_t^{MC}$, Ref. [66] noted that certain regions of the distributions are more

sensitive to $M_t^{MC}$ than others. In order to quantify the effect, Ref. [66] introduced a "local shape sensitivity" function defined by

$$\text{Local shape sensitivity} = \frac{1}{f(\boldsymbol{x}\,;\,\theta)}\left[\frac{\partial f(\boldsymbol{x}\,;\,\theta)}{\partial\theta}\right]^2, \tag{77}$$

which was also plotted in Figs. 1b, 3b, and 4b of [66]. In order to make the connection to our previous example, in Figure 10 we also plot with the blue dashed line the so-defined local shape sensitivity (77) (which in our language is $f u^2$). The fact that in both Figure 10 and the figures of Ref. [66], the distribution $f$ and the local shape sensitivity (77) have very different shapes demonstrates the scope for optimization by biasing the event generation appropriately. This optimization is precisely the motivation for the OASIS framework, and the results from Figure 6 and Figure 7 are indicative of the resource conservation which could be available to the LHC experimental collaborations, should such need arise.

## 4 Conclusions and outlook

In the high-luminosity era of the LHC, as well as in experiments at the intensity frontier, we expect to face a crunch for the computational resources required for Monte Carlo simulations. In this paper, we introduced a technique called OASIS for ameliorating this problem by modifying the sampling distribution used in the Monte Carlo. OASIS preferentially focuses the sampling of simulated events on certain regions of phase space (and appropriately weights them), in order to achieve the best experimental sensitivity for a given computational budget. We can also view the utility of OASIS as reducing the number of simulated events needed to reach a target sensitivity. We do this in two steps described in Section 3.2 and summarized with the flowcharts depicted in Figure 8 and Figure 9. In the remainder of this section, we will summarize and contextualize the key ideas of OASIS.

- *The use of* **weighted** *events in the samples representing the theory model predictions.* While the use of weighted events on an event-by-event basis is currently not very common in experimental analyses[18], the usefulness of the individual event weights was realised by theorists quite early on, and the event weight information was implemented in the Les Houches event file formats [71,72]. Recently, it has been pointed out that event weights can also be used to optimize the event selection and categorization in any given experimental analysis, leading to higher sensitivity [73,74]. The event weighting procedure is actually very straightforward—in fact, experimental analyses in some cases do effectively (re-)weight their events:

  - For example, when adding subsamples from different underlying processes with the same experimental signature, the individual subsamples have to be weighted appropriately to get their proportions right.
  - Another current use example is the oversampling of the tails of distributions, which can be done by either manually "slicing" with different generation cuts or by "biasing" the parton-level phase space with user-defined suppression factors [7]. While both of these approaches help reduce the overall event generation resource requirements, here we are proposing to exploit the idea to an unprecedented degree.

- *The choice of the sampling distribution should be dictated not only by the expected statistics, but also by the utility of a given event to the experimental analysis.* The standard approach (importance sampling) generates events according to their expected frequency

---

[18]MC reweighting was used by some LEP experiments in the late 1990s to measure both particle masses [67,68] and couplings [69], and more recently by CMS to set limits on anomalous vector boson couplings [70].

in the data. However, as we showed here, the most common events in the data are not necessarily the most useful in the information-theoretic sense. OASIS directly takes into account the sensitivity of different phase space regions to a parameter of interest as encoded by the per-event score (at the parton or at the analysis level, as desired). Thus the OASIS approach, by construction, seeks to maximize the utility of the simulated dataset.

- *Optimizing for unknown signal parameters.* One potential complication when applying OASIS to the MC generation of new physics samples is that a priori we do not know the values of the BSM parameters (even approximately). In that case, the standard procedure is to optimize the sensitivity for the parameter values which are close to the *expected* exclusion or discovery contours in parameter space. The same approach can be applied to OASIS as well.

- *Aligning the resource allocation with the established priorities of the experiment.* The main objective of any experimental analysis is achieving the highest possible sensitivity. By linking the choice of the sampling distribution to the per-event score values, OASIS aligns the goals of MC event generation and the physics analysis. For a single analysis, this is straightforward. However, the situation is complicated by the fact that modern experimental collaborations are quite large, comprised of many analysis groups, with potentially conflicting views on the relative importance of different phase space regions for their analyses. Nevertheless, reaching a consensus is certainly possible, as evidenced by the accepted agreements for the trigger menu, where different analysis groups are similarly competing for trigger bandwidth.

- *Increased coordination between the analysis and Monte Carlo generation teams within each experiment.* The successful implementation of OASIS also requires unprecedented level of cooperation and interaction among the MC generation and physics analysis groups within the experimental collaborations. The MC group will have to rely on the feedback from the analysts in designing the optimal sampling distribution.

- *Taking stock of current resource usage.* A very achievable near-term goal for any LHC collaboration would be to start tracking the current use of MC generated events, and identify the classes of events which tend to be used a lot (by many analysis groups) and conversely, the classes of events which, once produced, tend to be underutilized. This inventorization (which is in principle a very easy first step, since each simulated event carries a unique label) would go a long way towards implementing the general philosophy of OASIS in practice.

- *Optimizing for model exploration.* In this paper we showcased OASIS for measuring the parameters[19] of a theory model. However, the OASIS technique is also applicable when one is interested in quantifying how well the data fits the theory model. For example, if we can parametrize the possible deviations of interest from a reference model, the results from this paper still hold, albeit with respect to the parameters capturing said deviations.

Looking ahead, the HEP community has identified an ambitious and broad experimental program for the coming decades, which would require large investments not only in new facilities and experiments, but also in the R&D and computational resources for the associated software [4]. One aspect of this program is "improving the efficiency of event generation as used by the experiments", which was identified in [4] as an underexplored avenue in event

---

[19]Although in this paper we focused on the single parameter measurement case, our technique can be extended to work for the simultaneous measurement of several parameters.

generation R&D. OASIS directly addresses this by undersampling regions of the parton-level phase space which are less useful to the experimental analyses, thus realizing the goal set out in [4].

# Acknowledgements

The authors would like to thank D. Acosta, S. Gleyzer, S. Höche, S. Mrenna, D. Noonan, K. Pedro, and G. Perdue for useful discussions.

**Funding information**    The work of PS is supported in part by the University of Florida CLAS Dissertation Fellowship funded by the Charles Vincent and Heidi Cole McLaughlin Endowment. This work was supported in part by the United States Department of Energy under Grant No. DE-SC0010296.

# Code and data availability

The code and data that support the findings of this study are openly available at the following URL: https://gitlab.com/prasanthcakewalk/code-and-data-availability/ under the directory named arXiv_2006.16972.

# A  Fisher information for datasets with Poisson-distributed total number of events

In this section, we will derive the expression (7a) for the Fisher information contained in datasets whose total number of events is a Poisson-distributed random variable, with possibly $\theta$ dependent cross-sections. Recall that $f(\boldsymbol{x}\,;\,\theta)$ is the differential cross-section of $\boldsymbol{x}$ for a given value of $\theta$, $F(\theta)$ is the total cross-section, and $L$ is the integrated luminosity.

Let $k$ (a random variable) be the number of events in the dataset, and let $(\boldsymbol{x}_1,\ldots,\boldsymbol{x}_k)$ be the $\boldsymbol{x}$ values of the ordered collection of events.[20] Let $\Upsilon \equiv [k,(\boldsymbol{x}_1,\cdots,\boldsymbol{x}_k)]$ represent (an instance of) the dataset. The probability density of $\Upsilon$ is given by

$$\mathcal{P}(\Upsilon\,;\,\theta) = \frac{e^{-LF(\theta)}\,[LF(\theta)]^k}{k!}\,\prod_{i=1}^{k}\frac{f(\boldsymbol{x}_i\,;\,\theta)}{F(\theta)} \tag{78a}$$

$$= \frac{L^k\,e^{-LF(\theta)}}{k!}\,\prod_{i=1}^{k}f(\boldsymbol{x}_i\,;\,\theta)\,. \tag{78b}$$

Here we have used the fact that $k$ is Poisson-distributed with mean $LF(\theta)$, and the fact that the $\boldsymbol{x}_i$-s are independent of each other. This probability density is normalized as[21]

$$\int d\Upsilon\,\mathcal{P}(\Upsilon\,;\,\theta) \equiv \sum_{k=0}^{\infty}\int d\boldsymbol{x}_1\cdots\int \boldsymbol{x}_k\,\mathcal{P}(\Upsilon\,;\,\theta) = 1\,. \tag{79}$$

---

[20]Ordering of events can be derived from some form of event id. Ordering is demanded just so we do not have to worry about combinatorial factors when writing down the probability (density) of a particular instance of $\Upsilon$, since the events are distinguishable.

[21](79) implicitly specifies the reference measure with respect to which the probability density of $\Upsilon$ is defined.

From (78b), we get

$$\frac{\partial \mathcal{P}(\Upsilon ; \theta)}{\partial \theta} = \left[ -L \frac{\partial F(\theta)}{\partial \theta} + \sum_{i=1}^{k} \frac{\partial \ln f(\boldsymbol{x}_i ; \theta)}{\partial \theta} \right] \mathcal{P}(\Upsilon ; \theta) . \tag{80}$$

Next we derive (7a) starting from the expression for the Fisher information [56] in (81a) as follows (the explanations for the steps are provided below):

$$\mathcal{I}(\theta) = \int d\Upsilon \, \frac{1}{\mathcal{P}(\Upsilon ; \theta)} \left[ \frac{\partial \mathcal{P}(\Upsilon ; \theta)}{\partial \theta} \right]^2 \tag{81a}$$

$$= \int d\Upsilon \, \mathcal{P}(\Upsilon ; \theta) \left[ -L \frac{\partial F(\theta)}{\partial \theta} + \sum_{i=1}^{k} \partial_\theta \ln f(\boldsymbol{x}_i ; \theta) \right]^2 \tag{81b}$$

$$= E_{\mathcal{P}} \left[ \left[ -L \frac{\partial F(\theta)}{\partial \theta} + \sum_{i=1}^{k} \partial_\theta \ln f(\boldsymbol{x}_i ; \theta) \right]^2 \right] \tag{81c}$$

$$= L^2 \left[ \frac{\partial F(\theta)}{\partial \theta} \right]^2 - 2L \frac{\partial F(\theta)}{\partial \theta} E_{\mathcal{P}}[k ; \theta] E_f [\partial_\theta \ln f(\boldsymbol{x} ; \theta)]$$
$$+ E_{\mathcal{P}}[k ; \theta] E_f \left[ [\partial_\theta \ln f(\boldsymbol{x} ; \theta)]^2 \right]$$
$$+ E_{\mathcal{P}}[k^2 - k ; \theta] \left[ E_f [\partial_\theta \ln f(\boldsymbol{x} ; \theta)] \right]^2 \tag{81d}$$

$$= L^2 \left[ \frac{\partial F(\theta)}{\partial \theta} \right]^2 - 2L \frac{\partial F(\theta)}{\partial \theta} L F(\theta) \frac{1}{F(\theta)} \frac{\partial F(\theta)}{\partial \theta}$$
$$+ L F(\theta) \frac{1}{F(\theta)} \int d\boldsymbol{x} \, \frac{1}{f(\boldsymbol{x} ; \theta)} \left[ \frac{\partial f(\boldsymbol{x} ; \theta)}{\partial \theta} \right]^2$$
$$+ [L F(\theta)]^2 \left[ \frac{1}{F(\theta)} \frac{\partial F(\theta)}{\partial \theta} \right]^2 \tag{81e}$$

$$= L \int d\boldsymbol{x} \, \frac{1}{f(\boldsymbol{x} ; \theta)} \left[ \frac{\partial f(\boldsymbol{x} ; \theta)}{\partial \theta} \right]^2 , \tag{81f}$$

where $E_{\mathcal{P}}[\cdots]$ and $E_f[\cdots]$ represent the expectation values under $\mathcal{P}$ and $f$ respectively. Equation (81b) follows from plugging in (78b) and (80) in (81a). In (81c), we have used the definition of expectation value. In (81d), we have expanded the square in (81c) and used the fact that the $\boldsymbol{x}_i$-s are independent of each other. In (81e), we have used the following three results:

$$E_{\mathcal{P}}[k ; \theta] = L F(\theta) , \tag{82}$$

$$E_{\mathcal{P}}[k^2 ; \theta] = L F(\theta) + [L F(\theta)]^2 , \tag{83}$$

$$E_f [\partial_\theta \ln f(\boldsymbol{x} ; \theta)] = \frac{1}{F(\theta)} \int d\boldsymbol{x} \, f(\boldsymbol{x} ; \theta) \frac{1}{f(\boldsymbol{x} ; \theta)} \frac{\partial f(\boldsymbol{x} ; \theta)}{\partial \theta} \tag{84a}$$

$$= \frac{1}{F(\theta)} \frac{\partial F(\theta)}{\partial \theta} . \tag{84b}$$

Finally, by cancelling the first, second, and fourth additive terms in (81e), we get (81f) which matches (7a).

# B Optimal sampling in the $N_s \to \infty$ limit

In this section we will prove (28). Staring from (27b), we can write $\mathcal{I}_{\text{MC}}/L$ in the $N_s \to \infty$ limit as

$$\frac{\mathcal{I}_{\text{MC}}}{L} = \frac{I}{L} - \frac{L}{N_s} \int d\boldsymbol{x} \; g(\boldsymbol{x}) \, w^2(\boldsymbol{x}) \, u^2(\boldsymbol{x}) \tag{85a}$$

$$= \frac{I}{L} - \frac{L}{N_s} \int d\boldsymbol{x} \; g(\boldsymbol{x}) \frac{f^2(\boldsymbol{x}) \, u^2(\boldsymbol{x})}{g^2(\boldsymbol{x})} \tag{85b}$$

$$\leq \frac{I}{L} - \frac{L}{N_s} \left[ \int d\boldsymbol{x} \; g(\boldsymbol{x}) \frac{f(\boldsymbol{x}) \, |u(\boldsymbol{x})|}{g(\boldsymbol{x})} \right]^2 \tag{85c}$$

$$= \frac{I}{L} - \frac{L}{N_s} \left[ \int d\boldsymbol{x} \; f(\boldsymbol{x}) \, |u(\boldsymbol{x})| \right]^2 . \tag{85d}$$

In (85b) we have used $w = f/g$ and in (85c) we have used Jensen's inequality and the fact that the square function is convex. The result in (85d) sets an upper bound on $\mathcal{I}_{\text{MC}}$ (in the $N_s$ limit) that is independent of the sampling distribution $g$. This upper bound is reached when the equality in (85c) is reached, i.e., if $f(\boldsymbol{x})|u(\boldsymbol{x})|/g(\boldsymbol{x})$ is constant almost everywhere. This means that in the $N_s \to \infty$ limit, the optimal sampling distribution is proportional to $f(\boldsymbol{x})|u(\boldsymbol{x})|$, which leads to (28). We use the absolute value of $u$ in (85c) since $g(\boldsymbol{x})$ is constrained to be non-negative.

# C Dependence of the optimal weights on only $|u|$

In this section we will prove (44), i.e., that for any sampling distribution $g$ (with weights $w$), there exists a sampling distribution $g_{\text{better}}$ (with weights $w_{\text{better}}$) given by

$$g_{\text{better}}(\boldsymbol{x}) = \frac{f(\boldsymbol{x})}{E_g \left[ w(\boldsymbol{x}) \,\middle|\, |u(\boldsymbol{x})| \right]} , \tag{86}$$

$$w_{\text{better}}(\boldsymbol{x}) = \frac{f(\boldsymbol{x})}{g_{\text{better}}(\boldsymbol{x})} = E_g \left[ w(\boldsymbol{x}) \,\middle|\, |u(\boldsymbol{x})| \right] , \tag{87}$$

such that the $\mathcal{I}_{\text{MC}}$ under $g_{\text{better}}$ is greater than or equal to the $\mathcal{I}_{\text{MC}}$ under $g$.

We will begin by showing that $g_{\text{better}}$ is a unit-normalized distribution:

$$\int d\boldsymbol{x} \; g_{\text{better}}(\boldsymbol{x}) = E_g \left[ \frac{w}{w_{\text{better}}} \right] \tag{88a}$$

$$= E_g \left[ \frac{1}{w_{\text{better}}} E_g \left[ w \,\middle|\, |u| \right] \right] \tag{88b}$$

$$= E_g \left[ \frac{w_{\text{better}}}{w_{\text{better}}} \right] = 1 , \tag{88c}$$

where $E_g[\cdots]$ represents the expectation value under the sampling distribution $g$. In (88a), we have used the fact that $g \, w = g_{\text{better}} \, w_{\text{better}}$. In (88b), we have used the definition of conditional expectation and the fact that $w_{\text{better}}$ is fixed for a given value of $|u|$. In (88c), we have used the expression for $w_{\text{better}}$ from (87).

The proof of (44) proceeds as follows (the explanations for the steps are provided below):



$$\left\{ \frac{\mathcal{I}_{\mathrm{MC}}}{L} \text{ under } g \right\} = \int d\boldsymbol{x} \; f(\boldsymbol{x}) \; \frac{u^2(\boldsymbol{x})}{1 + \frac{L}{N_s} w(\boldsymbol{x})} = \int d\boldsymbol{x} \; g(\boldsymbol{x}) \; \frac{w(\boldsymbol{x}) u^2(\boldsymbol{x})}{1 + \frac{L}{N_s} w(\boldsymbol{x})} \tag{89a}$$

$$= E_g \left[ \frac{w \, u^2}{1 + L \, w/N_s} \right] \tag{89b}$$

$$= E_g \left[ u^2 \, E_g \left[ \frac{w}{1 + L \, w/N_s} \; \middle| \; |u| \right] \right] \tag{89c}$$

$$\leq E_g \left[ u^2 \, \frac{E_g \left[ w \, \middle| \, |u| \right]}{1 + (L/N_s) E_g \left[ w \, \middle| \, |u| \right]} \right] \tag{89d}$$

$$= E_g \left[ \frac{w_{\mathrm{better}} u^2}{1 + L \, w_{\mathrm{better}}/N_s} \right] \tag{89e}$$

$$= E_{g_{\mathrm{better}}} \left[ \frac{w_{\mathrm{better}}}{w} \, \frac{w_{\mathrm{better}} u^2}{1 + L \, w_{\mathrm{better}}/N_s} \right] \tag{89f}$$

$$= E_{g_{\mathrm{better}}} \left[ \frac{w_{\mathrm{better}}^2 u^2}{1 + L \, w_{\mathrm{better}}/N_s} \, E_{g_{\mathrm{better}}} \left[ w^{-1} \, \middle| \, |u| \right] \right] \tag{89g}$$

$$= E_{g_{\mathrm{better}}} \left[ \frac{w_{\mathrm{better}}^2 u^2}{1 + L \, w_{\mathrm{better}}/N_s} \, E_g \left[ w_{\mathrm{better}}^{-1} \, \middle| \, |u| \right] \right] \tag{89h}$$

$$= E_{g_{\mathrm{better}}} \left[ \frac{w_{\mathrm{better}} u^2}{1 + L \, w_{\mathrm{better}}/N_s} \right] \tag{89i}$$

$$= \left\{ \frac{\mathcal{I}_{\mathrm{MC}}}{L} \text{ under } g_{\mathrm{better}} \right\} , \tag{89j}$$

where $E_g[\cdots]$ and $E_{g_{\mathrm{better}}}[\cdots]$ represent the expectation values under the sampling distributions $g$ and $g_{\mathrm{better}}$ respectively.

In (89a), we have used the definitions of $\mathcal{I}_{\mathrm{MC}}$ and $w$. Equation (89b) follows from the definition of $E_g[\cdots]$. In (89c), we have used the fact that $u^2$ is uniquely defined for a given value of $|u|$. In (89d), we have used the conditional version of Jensen's inequality and the fact that $w/(1 + \alpha w)$ is a concave function in $w$ for a positive $\alpha$. In (89e), we have used the expression for $w_{\mathrm{better}}$ from (87). In (89f), we have used the fact that $g \, w = g_{\mathrm{better}} \, w_{\mathrm{better}}$. In (89g), we have used the fact that $u^2$ and $w_{\mathrm{better}}$ are fixed for a given value of $|u|$. In (89h), we used the fact that $f\left(\boldsymbol{x} \, \middle| \, |u|\right) = g\left(\boldsymbol{x} \, \middle| \, |u|\right) w = g_{\mathrm{better}}\left(\boldsymbol{x} \, \middle| \, |u|\right) w_{\mathrm{better}}$. In (89i), we have used the fact that $w_{\mathrm{better}}$ is fixed for a given value of $|u|$. The equality in (89j) is analogous to the equality in (89b).

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
