# Peer review of "OASIS: Optimal Analysis-Specific Importance Sampling for event generation"

_SciPost Physics, doi:SciPost Phys. 10, 034 (2021)_

## Round 1 · Referee Report · Anonymous · 2020-8-13

Strengths

1. covering both how the method can be applied at parton level and detector level analysis
2. paper well written and pedagogical
3. paper particularly relevant for the HL-HLC where such approach can be keys for the event-generation

Weaknesses

1. weak conclusion
2. missing systematical uncertainty impact in the parton-level analysis

Report

The paper focus on how someone can optimized the event generation for a given analysis in order to reduce as much as possible the statistical uncertainty impact on that given analysis. The main idea behind such method

The paper is well written and very pedagogical in the way they present the various concept used in the paper.

The paper also present two different type of analysis, the first one at parton level, which is particularly interesting for theorist and phenomenologist and the second one at reconstructed level which is particularly interesting for experimentalist. I would have been keen to recommend publication for each of the them independently.

The main drawback of the paper is that the search of optimal event generation for a given analysis might not lead to an optimal event set for another analysis. While this might not be a real concern for theorist/phenomenologist, this can be a huge issue for the experimental community where sample of events are primary produced centrally and the same events sample used by many group and an even higher number of analysis. This being said this is nothing that the authors can do about it, it just reduce the significance of their work to my point of view (and the issue is briefly mentioned in the paper already).

A final minor criticism is about the conclusion of the paper. My personal feeling is that the conclusion feels a bit too much like an introduction with the comparison of more basic method (for example biasing, slicing,...). Those comparison are valid but should be part of the introduction rather than the conclusion of the paper. The conclusion of the paper would also gain to recap in bit more in details how the optimality is defined/reached and some of the practical implication (like the fact that even for an infinite budget the optimal distribution is not the unweighted one).

As a final suggestion to the author, I believe that the paper can be significantly be improve by adding at the partonic level the effect of theoretical uncertainty. The authors include such type of effects in the detector level analysis while such effect are also important for the partonic case and seeing how figure 6 is impacted would be a nice plot to look at (even if I can guess what the plot will look likes in advance). But I think this can be left to the author to decide to include this or not.

Finally, I want to point to the author, a minor typo in the paper on page 23 -last line- the word "the" is repeated.

My final recommendation for this paper is a minor revision, such that the authors can at minima improve the conclusion and consider the other points that i have raised. After that small change, my recommendation will change to
"publish --top 10%--"

Requested changes

1. rewrite the conclusion
2. consider to add description of the impact of theoretical uncertainty at parton-level (optional)

  • validity: top
  • significance: good
  • originality: high
  • clarity: top
  • formatting: excellent
  • grammar: perfect

Author:  Prasanth Shyamsundar  on 2020-12-28  [id 1116]

(in reply to Report 1 on 2020-08-13)

Thank you for the constructive feedback! We have made the following requested changes:
1. We have rewritten and expanded the conclusions sections following the input from both referees, and feedback from our talks on this work. 2. In the original version, the theoretical uncertainty was already present in eq. (63) in the form of $\sigma_\mathrm{syst}$. In order to highlight this, we expanded the text above eq. (63), discussing the different contributions to $\sigma_\mathrm{syst}$.
Also, thanks for pointing out the repeated word---we've fixed it.

---

## Round 1 · Referee Report · Anonymous · 2020-8-21

Strengths

1. The paper is timely as analyzing the expected large data sets at HL-LHC will require very large Monte Carlo (MC) samples. Optimizing the use of computational resources to generate such samples is important to maximize physics potential of HL-LHC.

2. The paper proposes an elegant idea to achieve this task, and discusses it in a clear, transparent, and mathematically rigorous way.

3. The paper describes in detail how this idea can be implemented in realistic experimental conditions, and as an example points to a concrete CMS analysis that can clearly benefit from it.

4. The idea is broadly applicable, not just in collider physics but in any area where comparison of experimental data and MC is employed.

Weaknesses

1. A few questions regarding trade-offs inherent in the proposed method, and the range of analyses that can be benefit from it, have not yet been addressed. (See report for details.)

Report

1. In the example considered in Sec. 2, it is shown that the proposed method improves the measurement uncertainty of the parameter $\theta$ possible for fixed $N_s$. The uncertainty is effectively computed with the prior that the data is correctly described by the function $f$. In general, theory/data comparison needs to address two questions - how well the data fits the underlying theoretical model, and what are the theory parameter(s) that fit the data best. It seems to me that the proposed technique may optimize the power to address the second question at the expense of reducing the power to address the first, and that the optimization would look different (and probably closer to the usual goal of $g(x)=f(x)$) if both questions are optimized together.

2. In applications to searches for BSM physics, the "optimal" region of phase space usually strongly depends on the parameters of the underlying BSM model. (For example, in a bump hunt, one would want to focus on the invariant masses around the mass of the hypothetical new particle.) The proposed method would require different optimization for each assumed set of BSM parameters, and since they are typically not known a priori, it is not clear that an overall reduction in MC sample sizes could be achieved. (An exception may be a model with very specific predictions, or if a signal already seen in one channel is being searched for in other channels.)

3. On p. 29, the authors state that "the event variable $v$ for a given value of parton-level event $x$ typically fall[s] within a small window of possibilities". However in some cases we may need to model tails of distributions which may contain significant contributions from rare events with "highly atypical" $x \mapsto v$ maps, e.g. large mis-measurements of MET etc. Would the proposed method still apply and be useful?

Requested changes

The authors should consider the questions listed above and add discussion of these points to the paper as they see fit.

  • validity: top
  • significance: high
  • originality: high
  • clarity: top
  • formatting: perfect
  • grammar: perfect

Author:  Prasanth Shyamsundar  on 2020-12-28  [id 1115]

(in reply to Report 2 on 2020-08-21)

Thank you for the constructive feedback! We have made the following requested changes:
1. Thank you for raising this point. We have added a corresponding discussion in the conclusions section (under the bullet Optimizing for model exploration). To quickly answer the question here, even under those circumstances, our technique can give a non-trivial result, different from $g(x)=f(x)$. If the goal is to quantify how well the data fits the underlying theoretical model, our technique can help if we can parametrize the possible deviations from $f$ that one is interested in. In that case, the results of the paper can be used with respect to these additional parameters. If the goal is to maximize the significance of an unmodeled/unparameterized search, then the problem of finding the optimal sampling distribution is ill posed because we cannot predict ahead of time where the maximal deviations will occur. Therefore, no sampling distribution will be universally optimal in all new physics scenarios, but, as you pointed out, in this case the heuristic $g=f$ is a natural choice. 2. This is a valid point, although the same problem plagues any sampling scheme. What is being done currently in ATLAS and CMS is to focus on the parameter values which are close to the expected exclusion or discovery contours in parameter space. Our approach fits naturally in this scheme. We have added a corresponding discussion in the conclusions (under the bullet Optimizing for unknown signal parameters). 3. The short answer is yes. We have elaborated on this in the first and second bullets of section 3.2.2.

---

## Round 2 · Referee Report · Anonymous (Referee 2) · 2021-1-21

Report

The resubmitted version of this manuscript adequately addresses the questions raised in my original report. I recommend that the paper be published in SciPost Physics in its present form.

---

## Round 2 · Referee Report · Anonymous (Referee 1) · 2021-1-25

Report

The authors have improved their paper. I'm happy to recomend it for publication.

---

## Round 2 · List of Changes

1. The conclusions sections has been completely rewritten, to incorporate the feedback from both referees.
  2. For clarity of the presentation, we have expanded the notation and related definitions pertaining to equation 63 and onwards.
  3. In response to referee 2's query, we have expanded the first and second bullets in section 3.2.2.
  4. To address referee 1's suggestion, we have we expanded the text above eq. (63), discussing the different contributions to $\sigma_\mathrm{syst}$.

---

## Editorial Decision

published